# The Futile Creatine Cycle powers UCP1-independent thermogenesis in classical BAT

Jakub Bunk[1,2], Mohammed F. Hussain[1,2], Maria Delgado-Martin[1,2], Bozena Samborska[1], Mina Ersin[1,2], Abhirup Shaw ●[1], Janane F. Rahbani ●[1] & Lawrence Kazak ●[1,2] ✉

Classical brown adipose tissue (BAT) is traditionally viewed as relying exclusively on uncoupling protein 1 (UCP1) for thermogenesis via inducible proton leak. However, the physiological significance of UCP1-independent mechanisms linking substrate oxidation to ATP turnover in classical BAT has remained unclear. Here, we identify the Futile Creatine Cycle (FCC), a mitochondrial-localized energy-wasting pathway involving creatine phosphorylation by creatine kinase b (CKB) and phosphocreatine hydrolysis by tissue-nonspecific alkaline phosphatase (TNAP), as a key UCP1-independent thermogenic mechanism in classical BAT. Reintroducing mitochondrial-targeted CKB exclusively into interscapular brown adipocytes in vivo restores thermogenesis and cold tolerance in mice lacking native UCP1 and CKB, in a TNAP-dependent manner. Furthermore, mice with inducible adipocyte-specific co-deletion of TNAP and UCP1 exhibit severe cold-intolerance. These findings challenge the view that BAT thermogenesis depends solely on UCP1 because of insufficient ATP synthase activity and establishes the FCC as a physiologically relevant thermogenic pathway in classical BAT.

Non-shivering thermogenesis is predominantly localized to brown adipose tissue (BAT) and is critical for thermoregulation in a cold environment (when external temperatures fall below thermoneutrality)[1–3]. In addition to playing a crucial role in maintaining body temperature in rodents and human infants[4–8], the abundance of BAT is associated with cardiometabolic health in rodents and humans[9–12], and investigating adipose tissue thermogenesis in adult humans is currently a research focal point.

Thermogenesis occurs through increased oxidation of macronutrients, triggered by proton re-entry into the mitochondrial matrix which alleviates thermodynamic pressure on the electron transport chain. This process involves direct proton leakage from the mitochondrial intermembrane space to the matrix by uncoupling protein 1 (UCP1)[13] or coupling macronutrient oxidation to ATP synthesis to support futile cycles that enhance UCP1-independent thermogenesis through accelerated ATP turnover[14,15]. Over half a century ago, thermogenic adipocytes were proposed to utilize a combination of ATP

turnover and proton leak for thermogenesis[16]. However, the physiological role of ATP-linked thermogenesis in brown adipocytes has been questioned due to a lower stoichiometry between ATP synthase and the electron transport chain compared to tissues like liver and heart[17–19]. Consequently, the concept that classical BAT relies exclusively on proton leak via UCP1 for thermogenesis has remained axiomatic, leaving the physiological relevance of ATP-linked thermogenesis unexplored in vivo.

Creatine kinases facilitate reversible phosphoryl group transfer from adenosine triphosphate (ATP) to creatine and are encoded by four genes: muscle-type (CKM), brain-type (CKB), ubiquitous-type, mitochondrial (CKMT1), and sarcomeric-type, mitochondrial (CKMT2)[20,21]. This enzyme's compartmentalization at sites of ATP production and consumption forms the Phosphocreatine Circuit, crucial for maintaining the free energy of ATP hydrolysis[22–24]. This circuit is particularly vital in excitable cells like myocytes, neurons, and spermatozoa[25]. Since the creatine kinases that participate in the

[1]Rosalind & Morris Goodman Cancer Institute, McGill University, Montreal, QC, Canada. [2]Department of Biochemistry, McGill University, Montreal, QC, Canada. ✉e-mail: lawrence.kazak@mcgill.ca

Phosphocreatine Circuit consume and restore ATP with phosphocreatine as an intermediate, this pathway does not dissipate energy. In contrast, thermogenic adipocytes (beige and brown) utilize the Futile Creatine Cycle (FCC) for ATP-linked thermogenesis, accelerating ATP turnover through creatine phosphorylation and phosphocreatine hydrolysis[14,15,26–29].

Studies utilizing purified beige fat mitochondria initially observed a bioenergetic phenomenon where creatine triggered a super-stoichiometric release of ADP[14,30]. Subsequent studies involved mice genetically depleted of creatine in adipocytes, demonstrating reduced thermogenesis[26,27,31]. Biochemical and genetic studies in brown and beige fat mitochondria, cells and mice identified creatine kinase b (CKB)[15] and tissue-nonspecific alkaline phosphatase (TNAP)[29] as key mediators of the FCC where they function in tandem to trigger creatine phosphorylation and phosphocreatine hydrolysis, respectively to consume ATP and promote thermogenesis[14,15,28]. Mice genetically lacking *Ckb* or *Alpl* (the gene encoding TNAP) in fat exhibit metabolic derangements such as susceptibility to diet-induced obesity and impaired glucose homeostasis[15,29]. Recent work in mice has shown that CKB functions in parallel to the action of UCP1 to support body

temperature maintenance in the cold[32], underscoring the physiological importance of CKB-dependent thermogenesis. However, the manner through which CKB enhances thermogenesis, whether by mediating an energy dissipating FCC[15,28,29] or by supporting the general health of adipocytes, needs further clarification. Here, we show that classical brown adipocytes support UCP1-independent thermogenesis, driven by ATP turnover, through the FCC.

## Results

### Selective protein expression in brown adipocytes in vivo

To achieve selective expression of proteins in parenchymal brown adipocytes in vivo, we employed an adeno-associated virus (AAV) system utilizing the AAV-FLEX construct where protein expression is under the control of Cre/LoxP recombination (a modified Flex switch) and a chicken β-actin promoter with cytomegalovirus enhancer[33] (Fig. 1a). We delivered AAV by subcutaneous injection directly above the interscapular brown adipose tissue (iBAT) (Fig. 1a). We opted for this approach over surgically exposing and injecting iBAT directly because it is less invasive and avoids introducing additional variables that could potentially, on their own, impair BAT function and

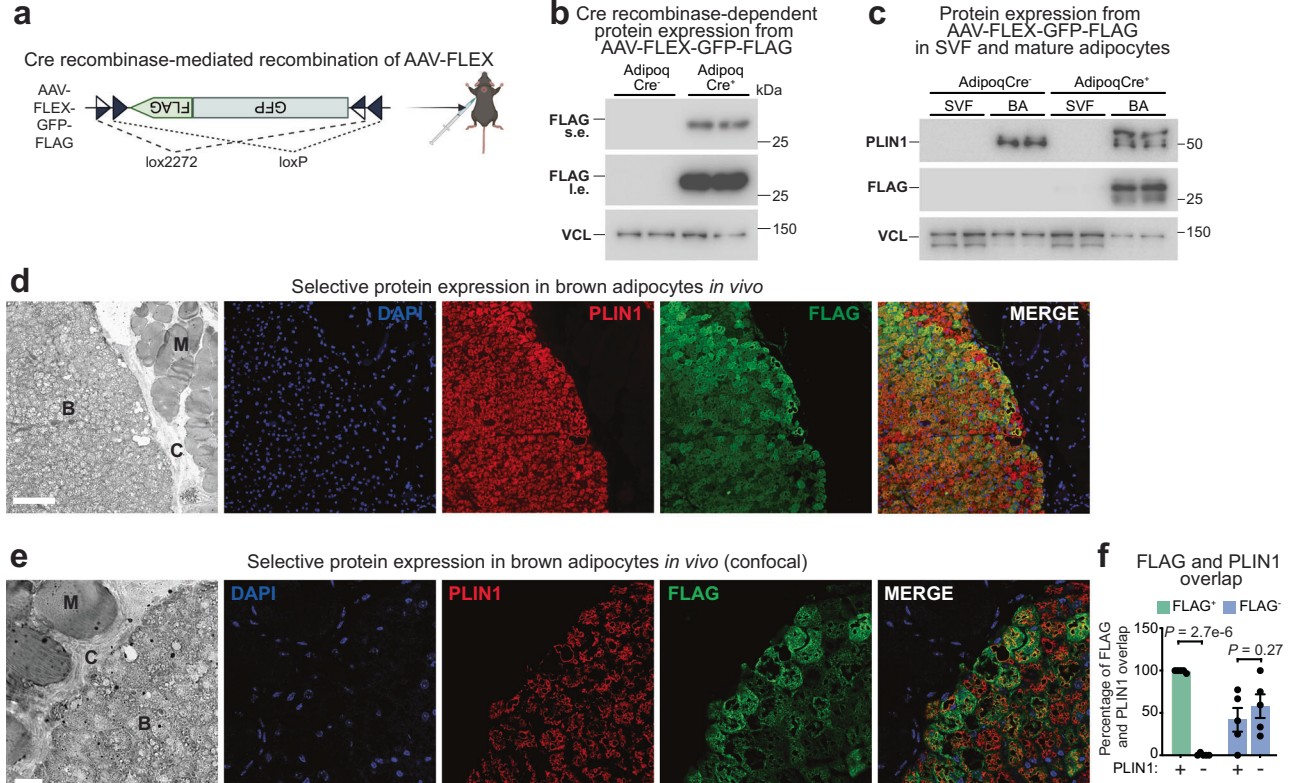

**Fig. 1 | Selective protein expression in brown adipocytes in vivo. a** Schematic of Cre recombinase-dependent AAV-FLEX recombination and interscapular brown adipose tissue (iBAT) transduction. Created in BioRender. Kazak, L. (2025) https://BioRender.com/j99d203. **b** Western blot of iBAT harvested from female AdipoqCre- and AdipoqCre+ mice (8 weeks of age), transduced with AAV-FLEX-GFP-FLAG (n = 2 per group). s.e., short exposure; l.e., long exposure. **c** Western blot of stromal vascular fraction (SVF) and acutely purified brown adipocytes (BA) from iBAT, harvested from female AdipoqCre- and AdipoqCre+ mice (16 weeks of age), one week after being transduced with AAV-FLEX-GFP-FLAG (n = 2 per group). **d** Representative immunofluorescence images of iBAT from female AdipoqCre+ mice transduced with AAV-FLEX-GFP-FLAG. Mature adipocytes were labelled with anti-Perilipin 1 (PLIN1) antibody (red), GFP-FLAG was labeled with anti-FLAG antibody (green). Nuclei were labelled with DAPI (blue). Brown adipocytes (B),

connective tissue (C), and muscle (M). Scale bars, 100 μm. This experiment was repeated 5 times with similar results. **e** Representative confocal immunofluorescence images of iBAT from female AdipoqCre+ mice transduced with AAV-FLEX-GFP-FLAG. Mature adipocytes were labelled with anti-Perilipin 1 (PLIN1) antibody (red), GFP-FLAG was labeled with anti-FLAG antibody (green). Nuclei were labelled with DAPI (blue). Brown adipocytes (B), connective tissue (C), and muscle (M). Scale bars, 20 μm. **f** The percentages of cells (total of 388 cells) that were positive or negative for FLAG (297 positive and 91 negative cells) and PLIN1 (328 positive and 60 negative cells) were counted manually from Fig. 1e and Supplementary Fig. 1a. f. Two-way ANOVA (Fisher's LSD test). Data are presented as the mean ± s.e.m. n numbers are of biologically independent experiments. Source data are provided as a Source Data file.

thermoregulation[34–36]. Although we recognized the importance of a less invasive approach for our cold exposure studies, we employed direct AAV injections into surgically exposed iBAT for other experiments, as detailed in a later section. Transduction with AAV-FLEX-GFP-FLAG (herein referred to as GFP) demonstrated selective GFP expression in AdipoqCre+ mice, but not AdipoqCre- mice (Fig. 1b). To further evaluate the adipocyte selectivity of Cre recombinase-driven over-expression, we compared GFP expression between mature brown adipocytes and the stromal vascular fraction (SVF) from iBAT, observing that the FLAG signal was highly enriched in mature adipocytes and nearly absent in the SVF fraction (Fig. 1c). Next, iBAT was subjected to immunofluorescence imaging to ascertain the specificity of protein expression within the interscapular region. Importantly, GFP expression was exclusively localized to perilipin 1-positive (PLIN1+) parenchymal brown adipocytes, with adjacent skeletal muscle and connective tissue near brown adipocytes showing no GFP expression (Fig. 1d, e). Quantification of confocal microscopy data (Fig. 1e and Supplementary Fig. 1a) demonstrated that 99% of FLAG+ cells also expressed PLIN1, with less than 1% of FLAG+ cells lacking PLIN1 expression (Fig. 1f). In contrast, FLAG- cells showed no significant enrichment for either PLIN1+ cells or PLIN1- cells (Fig. 1f). Thus, we conclude that AAV-FLEX-mediated protein synthesis is dependent on Cre recombinase activity, facilitating precise protein expression selectively in parenchymal brown adipocytes in vivo.

## Spatiotemporal protein expression in brown adipocytes in vivo

The FCC involves the coordinated action of CKB and TNAP, both localized within mitochondria, to facilitate ATP turnover via creatine phosphorylation and phosphocreatine hydrolysis (Fig. 2a). However, the expression of CKB outside mitochondria[15] suggests the presence of the Phosphocreatine Circuit in brown adipocytes (Fig. 2a). A requirement for CKB in cold-induced thermogenesis has been demonstrated by employing an inducible knockout approach[32]. To investigate the contribution of the FCC to thermogenesis, we utilized mice with pan-adipocyte inducible genetic deletion of Ckb and Ucp1 (iADKO^Ckb;Ucp1 mice)[32] to assess whether spatiotemporal protein expression from AAV-FLEX-transduced iBAT can be synchronized with endogenous Ckb and Ucp1 inactivation in mature adipocytes. The iBAT of iADKO^Ckb;Ucp1 mice was transduced with AAV-FLEX constructs encoding GFP-FLAG or CKB-FLAG (herein referred to as GFP or CKB, respectively), alongside constructs wherein the first 71 amino acids of the native inter membrane space (IMS) serine β-lactamase-like protein (LACTB)[37,38] was fused to the amino-terminus of CKB (herein referred to as L-CKB) or GFP (herein referred to as L-GFP), serving as control (Fig. 2b). Protein expression was induced by three consecutive daily intraperitoneal injections of tamoxifen (Fig. 2c). Robust expression of all variants was confirmed in tamoxifen-injected AdipoqCreERT2 mice (Fig. 2d) aligning with endogenous CKB and UCP1 protein ablation when iBAT of iADKO^Ckb;Ucp1 mice was transduced (Supplementary Fig. 1b, c). Western blot analysis revealed two distinct bands for L-GFP and L-CKB (Fig. 2d). Since the mature form of LACTB is generated by cleaving 62 amino acids from the leader sequence of the preprotein at the outer surface of the mitochondrial inner membrane[38], the slow and fast migrating bands (Fig. 2d and Supplementary Fig. 1c) likely represent the uncleaved and cleaved LACTB-fusions, respectively. Next, we examined the mitochondrial localization of our AAV-FLEX variants in iBAT using stimulated emission depletion (STED) microscopy. As expected, GFP did not localize to the mitochondria (marked by TOM20), whereas CKB exhibited partial mitochondrial colocalization, consistent with our previous findings[15] (Fig. 2e). Importantly, fusing LACTB to GFP or CKB increased their colocalization with TOM20 in iBAT (Fig. 2e). Next, we used differential centrifugation to purify mitochondria from the iBAT of iADKO^Ckb;Ucp1 mice that had been transduced with L-GFP or L-CKB. The mitochondrial proteins, Lon protease (LONP1) and mitochondrial heat shock protein 60 (HSP60), were enriched in

mitochondrial compared to whole tissue and cytosolic fractions (Fig. 2f). In addition, TNAP was enriched in mitochondria, consistent with its prior identification in these organelles in iBAT[29]. Notably, both L-GFP and L-CKB were strongly enriched in mitochondria compared to whole tissue and cytosolic fractions (Fig. 2f). Furthermore, creatine kinase activity was elevated exclusively in mitochondria of iBAT expressing L-CKB compared to L-GFP, while cytosolic creatine kinase activity remained unchanged (Fig. 2g). Subsequent western blotting and immunofluorescence analyses confirmed exclusive expression of L-CKB in PLIN1+ cells of iBAT in AdipoqCreERT2 mice, with no significant expression in PLIN1+ cells of subcutaneous or perigonadal white adipose tissues (Fig. 3a, b). Together these data show that AAV-FLEX-mediated protein synthesis is dependent on Cre recombinase activity, enabling spatiotemporal control of protein expression selectively in parenchymal brown adipocytes in vivo. Moreover, the ability to restrict AAV-FLEX-mediated protein expression to mitochondria of iBAT set the foundation for studying the contribution of the FCC to thermogenesis.

## Body temperature maintenance by the FCC in classical BAT

Due to the parallel thermogenic function of CKB and UCP1, iADKO^Ckb;Ucp1 mice display a cold-intolerant phenotype that is exacerbated compared to mice with individual Ckb or Ucp1 deletion[32]. We have previously shown that the energy expenditure in the cold (5 °C) rapidly increases to a similar level between cold-sensitive and cold-resistant mice[32]. However, the whole-body energy expenditure of mice that cannot maintain euthermia under these conditions drops because of an impaired ability to maintain body temperature. First, we sought to test whether the inability of iADKO^Ckb;Ucp1 mice to maintain euthermia in the cold was reproducible when protein expression from AAV-FLEX-transduced iBAT was concurrently triggered. We injected AAV-FLEX-LACTB-GFP-FLAG into the iBAT of female and male iADKO^Ckb;Ucp1 mice acclimated to room temperature (23 ± 1 °C). Three days later, we induced L-GFP expression and co-deleted Ckb and Ucp1 by administering tamoxifen intraperitoneally. Mice were then allowed to recover at 23 ± 1 °C for 4 days followed by 7–9 days at 30 °C before cold exposure (Fig. 4a). We confirmed successful deletion of native CKB and UCP1 proteins alongside AAV-mediated L-GFP protein expression (Fig. 4b). Our prior work demonstrated that when energy expenditure, as measured by indirect calorimetry, declines to 0.2 kcal/h at 5 °C, the rectal temperature of mice is lower than 28 °C, which is incompatible with survival in the cold. Hence, an energy expenditure level of 0.2 kcal/h was a criterium for early removal from cold challenge. Notably, nearly all iADKO^Ckb;Ucp1 mice expressing L-GFP were cold sensitive, with 94% requiring early removal within 10 h of exposure to 5 °C (Fig. 4c). Although male mice exhibited a higher maximal rise in energy expenditure compared to females (females: 0.51 ± 0.02 kcal/h; males: 0.66 ± 0.03 kcal/h), which delayed their removal from the cold (females: 5.25 h; males: 9.63 h), both sexes became hypothermic (Fig. 4c). These findings confirm our previous results[32], demonstrating the parallel requirement of CKB and UCP1 for cold-induced thermogenesis. Moreover, our data suggest that no other thermogenic pathway, beyond those affected by CKB and UCP1 ablation, provides adequate thermogenesis to sustain euthermia under cold conditions, at least under the experimental conditions we have utilized.

Next, we assessed body temperature regulation in response to cold in mice expressing either L-GFP or L-CKB in iBAT. Prior to cold exposure, mice housed at 30 °C exhibited comparable energy expenditure between groups (Supplementary Fig. 2a), with no differences in body weight before or after tamoxifen treatment (Supplementary Fig. 2b, c). In contrast to the high incidence of hypothermia observed in iADKO^Ckb;Ucp1 mice expressing L-GFP (Fig. 4c), expression of L-CKB greatly increased survival rates in both sexes (Fig. 4d, e). Among female iADKO^Ckb;Ucp1 mice expressing L-CKB, 22% survived 24 h in the cold (Fig. 4d) compared to 0% of L-GFP-expressing female mice (Fig. 4c).

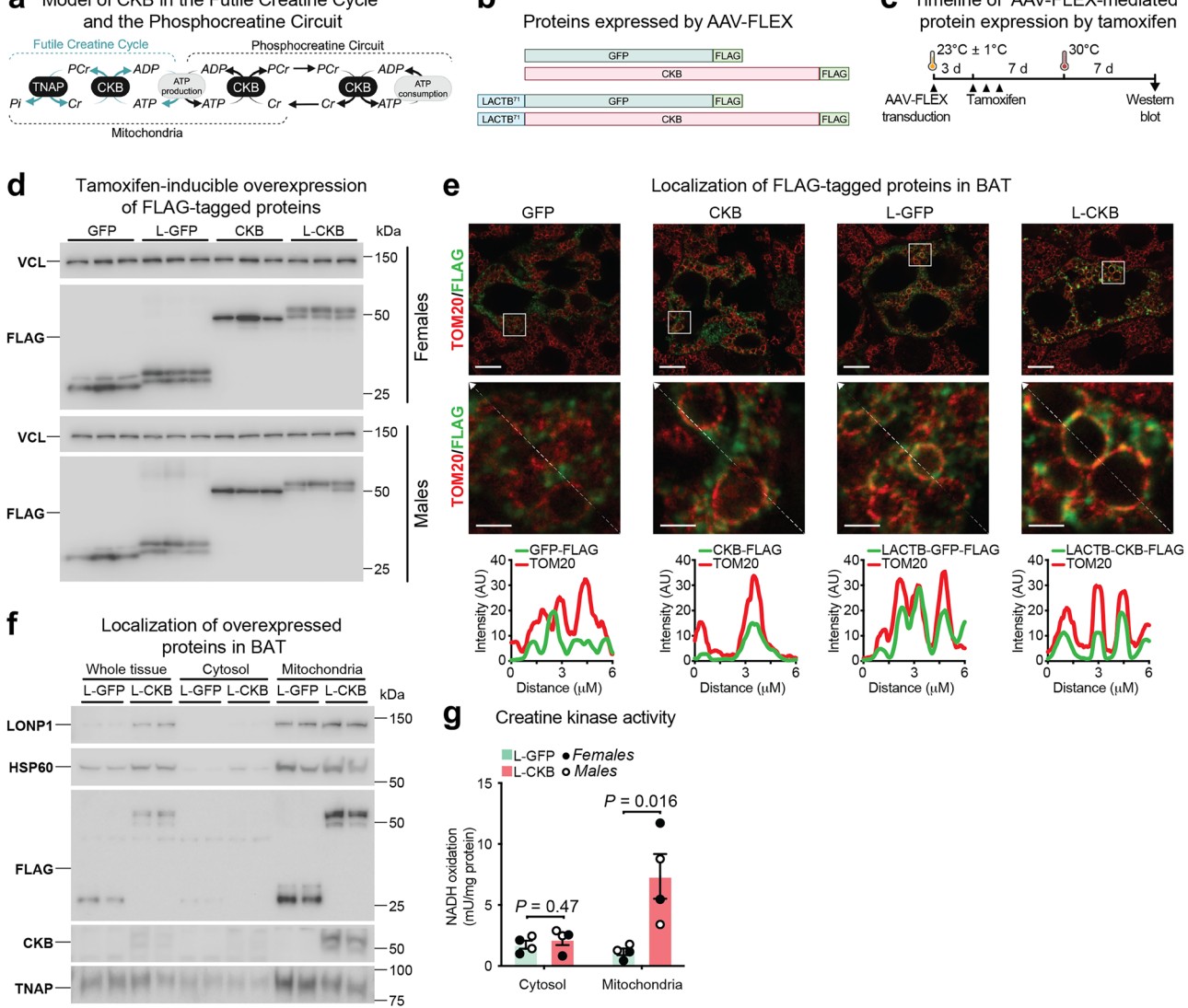

**Fig. 2 | Spatiotemporal protein expression in brown adipocytes in vivo.**
**a** Schematic of the Futile Creatine Cycle and the Phosphocreatine Circuit. In the Futile Creatine Cycle CKB and TNAP (both localized to mitochondria) work in tandem to phosphorylate creatine (Cr) and hydrolyze phosphocreatine (PCr) to sustain ATP turnover. In the Phosphocreatine Circuit, mitochondrial CKB supports the generation of PCr, which is then used by non-mitochondrial CKB to maintain the free energy of ATP hydrolysis. **b** Schematic of cDNAs cloned into pAAV-CA-FLEX for Cre recombinase-mediated overexpression. GFP, green fluorescent protein; CKB, creatine kinase b; LACTB[71], first 71 amino acids of LACTB that contain the leader sequence targeting it to mitochondria. **c** Schematic of AAV-FLEX transduction and tamoxifen-inducible protein expression and simultaneous *Ckb* and *Ucp1* co-deletion. Created in BioRender. Kazak, L. (2025) https://BioRender.com/j99d203. **d** Western blot of iBAT harvested from female and male iADKO[Ckb;Ucp1] mice (6-8 weeks of age), transduced with AAV-FLEXs expressing GFP-FLAG (GFP), LACTB-GFP-FLAG (L-GFP), CKB-FLAG (CKB), or LACTB-CKB-FLAG (L-CKB) (*n* = 3 per group). **e** Representative super-resolution (STED) microscopy

immunofluorescence images of iBAT from iADKO[Ckb;Ucp1] male mice transduced with AAV-FLEXs expressing GFP-FLAG (GFP), LACTB-GFP-FLAG (L-GFP), CKB-FLAG (CKB), or LACTB-CKB-FLAG (L-CKB). Mitochondria were labelled with anti-TOM20 antibody (red) and overexpressed proteins were labeled with anti-FLAG antibody (green). Scale bars, 5 µm. White box indicates the region depicting the zoomed image below. Scale bars, 1 µm. Relative TOM20 and FLAG signal intensity measured along the diagonal dashed arrow, expressed in arbitrary unites (AU), is shown below the immunofluorescence images. This experiment was repeated 3 times with similar results. **f** Western blot of whole tissue, cytosol, and mitochondrial fractions from iBAT. iBAT was harvested from L-GFP or L-CKB expressing female (*n* = 5, 7) and male (*n* = 6 per group) iADKO[Ckb;Ucp1] mice (6–8 weeks of age). **g** Creatine kinase activity of cytosolic and mitochondrial fractions isolated from iBAT of 9- to 11-week-old female and male iADKO[Ckb;Ucp1] mice (*n* = 2 per sex). Data are presented as mean ± s.e.m. *n* numbers are of biologically independent experiments. **g** Multiple two-sided unpaired *t*-test (Holm-Šídák correction). Source data are provided as a Source Data file.

Similarly, male iADKO[Ckb;Ucp1] mice showed an increase in 24-hour survival from 6% with L-GFP (Fig. 4c) to 26% with L-CKB (Fig. 4e). Notably, we observed variable levels of rescued L-CKB protein expression in iBAT among mice (Fig. 4f), which correlated significantly with cold survival time in both females and males (Fig. 4g, h). In contrast, L-GFP protein expression did not correlate with survival time in the cold (Supplementary Fig. 3a). Using a midpoint cutoff between minimal and maximal L-CKB expression within each sex, we categorized mice as "partial" or "complete" expressors (Fig. 4f–h). Grouping the energy

expenditure data based on these categories clearly differentiated survival outcomes in both sexes (Fig. 4i, j), demonstrating that complete L-CKB expression substantially prolonged cold survival compared to partial expression (Fig. 4k, l). Endogenous CKB and UCP1 were uniformly ablated regardless of variations in AAV-FLEX-derived protein levels (Fig. 4f) and were not associated with survival in the cold (Supplementary Fig. 4a–d). Conversely, L-CKB expression strongly correlated with enhanced cold resistance in the same samples (Supplementary Fig. 4e, f). In summary, our findings underscore that

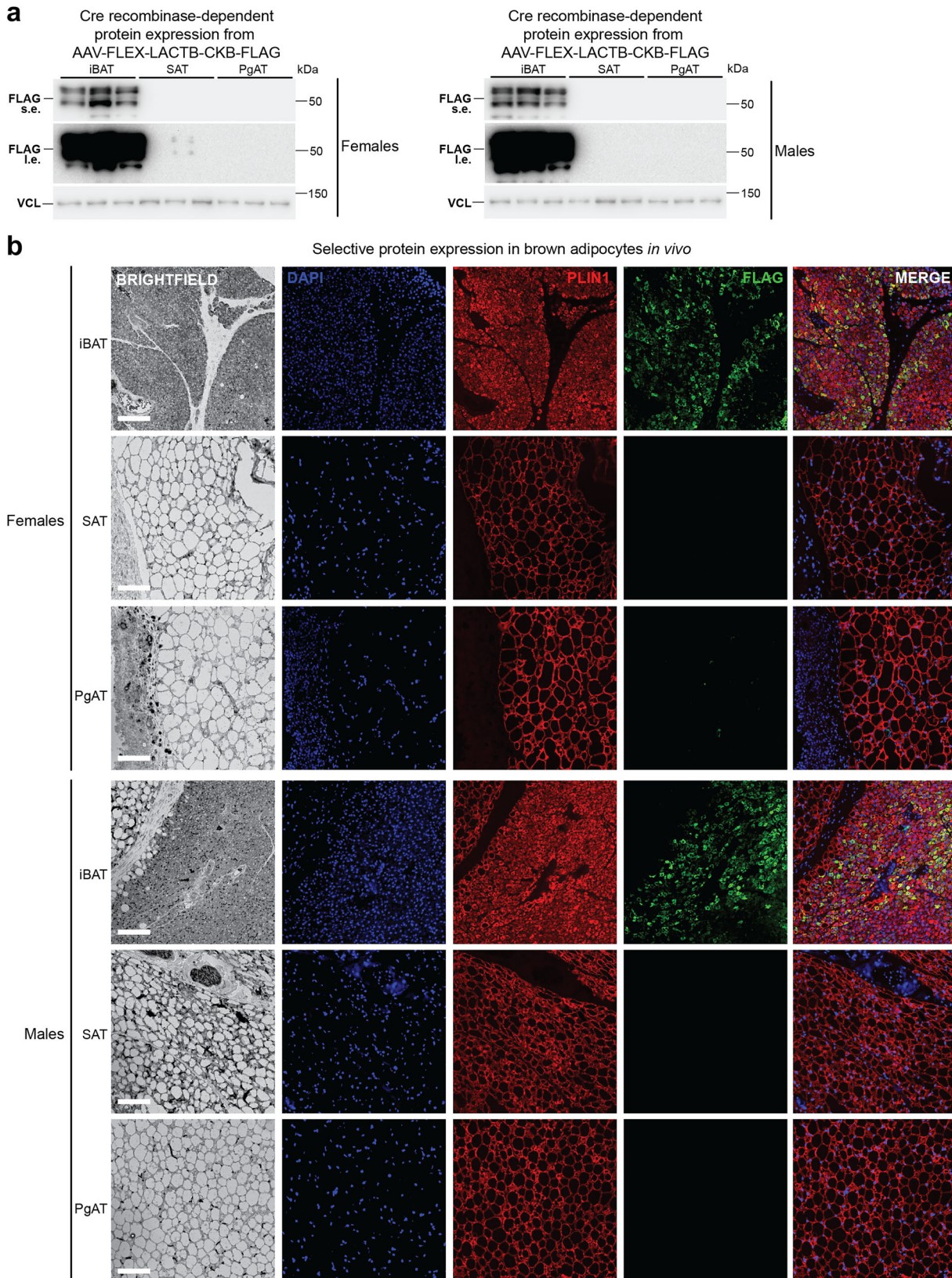

**Fig. 3 | AAV-FLEX-mediated protein expression in brown, not white, adipocytes in vivo. a** Western blot of interscapular brown adipose (iBAT), subcutaneous adipose tissue (SAT) and perigonadal adipose tissue (PgAT), of female and male iADKO^Ckb;Ucp1 mice (6–8 weeks of age) transduced with AAV-FLEX-LACTB-CKB-FLAG subcutaneously above the iBAT (*n* = 3 per group). **b** Representative immunofluorescence images of iBAT, SAT, and PgAT from female and male iADKO^Ckb;Ucp1 mice (6–8 weeks of age) transduced with AAV-FLEX-LACTB-CKB-FLAG subcutaneously above the iBAT. Mature adipocytes were labelled with anti-Perlipin 1 (PLIN1) antibody (red), GFP-FLAG was labeled with anti-FLAG antibody (green). Nuclei were labelled with DAPI (blue). Scale bars, 100 μm. This experiment was repeated 2 times for each sex with similar results. *n* numbers are of biologically independent experiments. Source data are provided as a Source Data file.

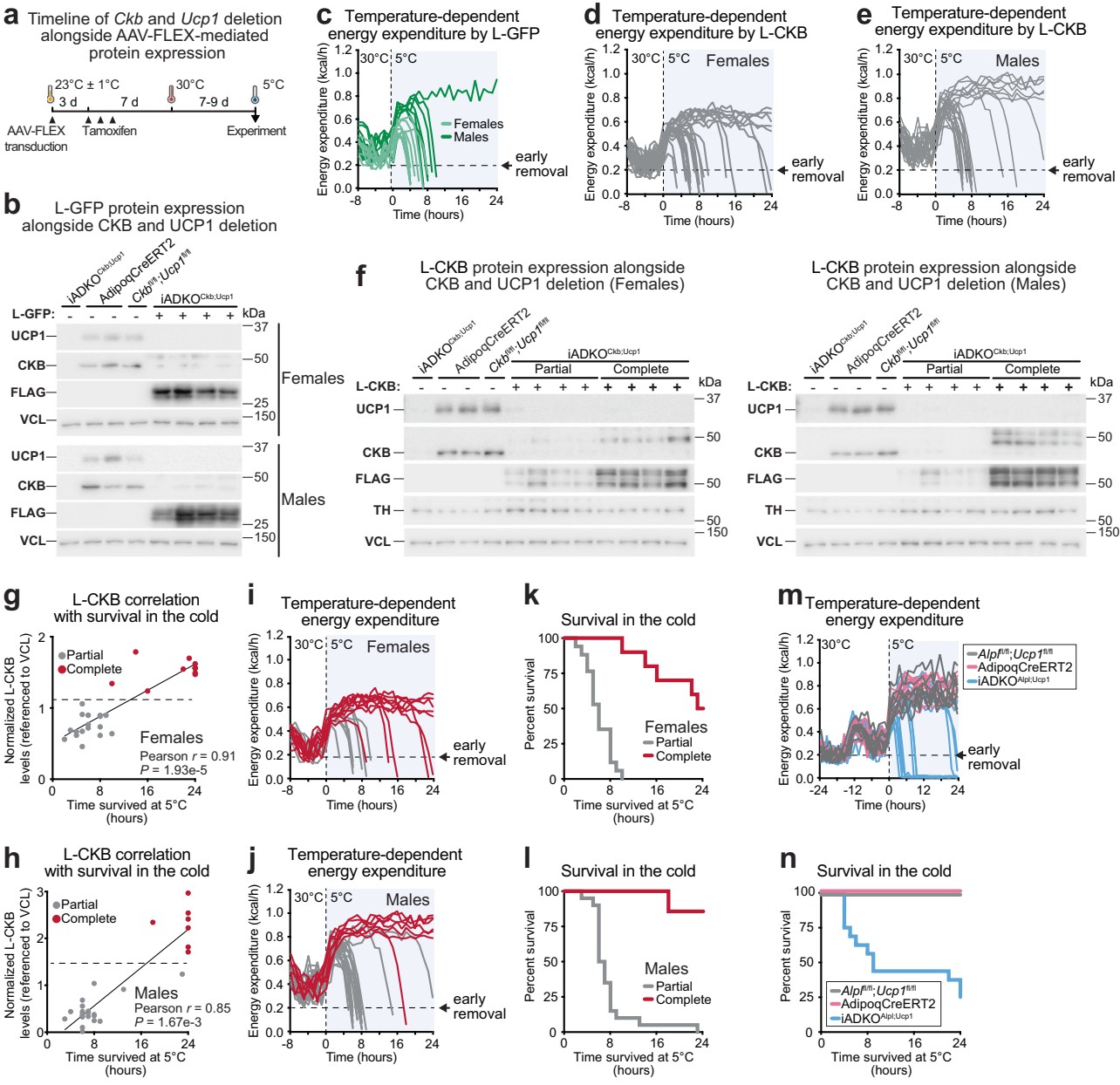

**Fig. 4 | Body temperature maintenance by the Futile Creatine Cycle in classical BAT. a** Schematic of AAV-FLEX transduction and tamoxifen-inducible protein expression in iADKO[Ckb;Ucp1] mice. Created in BioRender. Kazak, L. (2025) https://BioRender.com/j99d203. **b** Western blot of iBAT from 6- to 8-week-old iADKO[Ckb;Ucp1] mice transduced with AAV-FLEX-LACTB-GFP-FLAG (n = 4 per group). AdipoqCreERT2 and Ckb[fl/fl];Ucp1[fl/fl] mice were used as controls for tamoxifen-inducible deletion of endogenous UCP1 and CKB. **c** Energy expenditure (EE) at 30 °C and during acute 5 °C cold challenge of 8- to 13-week-old iADKO[Ckb;Ucp1] mice (n = 8 per sex) transduced subcutaneously into the iBAT depot with 1.5 × 10[11] viral infectious units of AAV-FLEX-LACTB-GFP-FLAG. Dashed line at 0.2 kcal/h marks early removal. EE at 30 °C and acute cold (5 °C) challenge of 8- to 13-week-old iADKO[Ckb;Ucp1] (**d**) female mice (n = 27) or (**e**) male mice (n = 27) transduced subcutaneously into the iBAT depot with 1.5 × 10[11] viral infectious units of AAV-FLEX-LACTB-CKB-FLAG. Dashed line at 0.2 kcal/h marks early removal. **f** Representative western blot of iBAT from iADKO[Ckb;Ucp1] mice expressing

partial and complete levels of L-CKB (n = 4 per group). AdipoqCreERT2 and Ckb[fl/fl];Ucp1[fl/fl] mice were used as controls for tamoxifen-inducible deletion of endogenous UCP1 and CKB. Pearson correlation of L-CKB protein expression and survival time at 5 °C in (**g**) female mice and (**h**) male mice (from experiments shown in **d** and **e**). **i**–**j** EE of (**i**) female mice and (**j**) male mice (from **d** and **e**) after assigning groups based on complete and partial L-CKB protein expression (from **g** and **h**). Kaplan Meier curve of (**k**) female mice and (**l**) male mice showing survival time at 5 °C (from **d** and **e**) after assigning groups based on L-CKB protein expression (from **g** and **h**). **m** EE at 30 °C and acute 5 °C challenge of 10- to 11-week-old female or male iADKO[Alpl;Ucp1], AdipoqCreERT2, and A1pl[fl/fl];Ucp1[fl/fl] mice (n = 8, 4, 4 per sex). Dashed line at 0.2 kcal/h marks early removal. **n** Kaplan Meier curve of Fig. 4m. **g**, **h** Pearson correlation (two-sided). Data are presented as the mean ± s.e.m. and n numbers are of biologically independent experiments. Source data are provided as a Source Data file.

mitochondrial-localized CKB in iBAT is sufficient to rescue the thermogenic impairment of iADKO[Ckb;Ucp1] mice, suggesting that the FCC in classical BAT plays a physiologically relevant role in body temperature maintenance. To further test the relevance of the FCC in cold-induced thermogenesis, we generated mice in which all adipocytes lacked both

Alpl and Ucp1 (iADKO[Alpl;Ucp1]) in an inducible manner. (Supplementary Fig. 4g). Strikingly, iADKO[Alpl;Ucp1] mice were profoundly more cold sensitive (75%) than either control (AdipoqCreERT2, or Alpl[fl/fl];Ucp1[fl/fl]) mice (Fig. 4m, n). These mice also displayed an exacerbated cold-sensitive phenotype in relation to single Ucp1 knockout mice, but

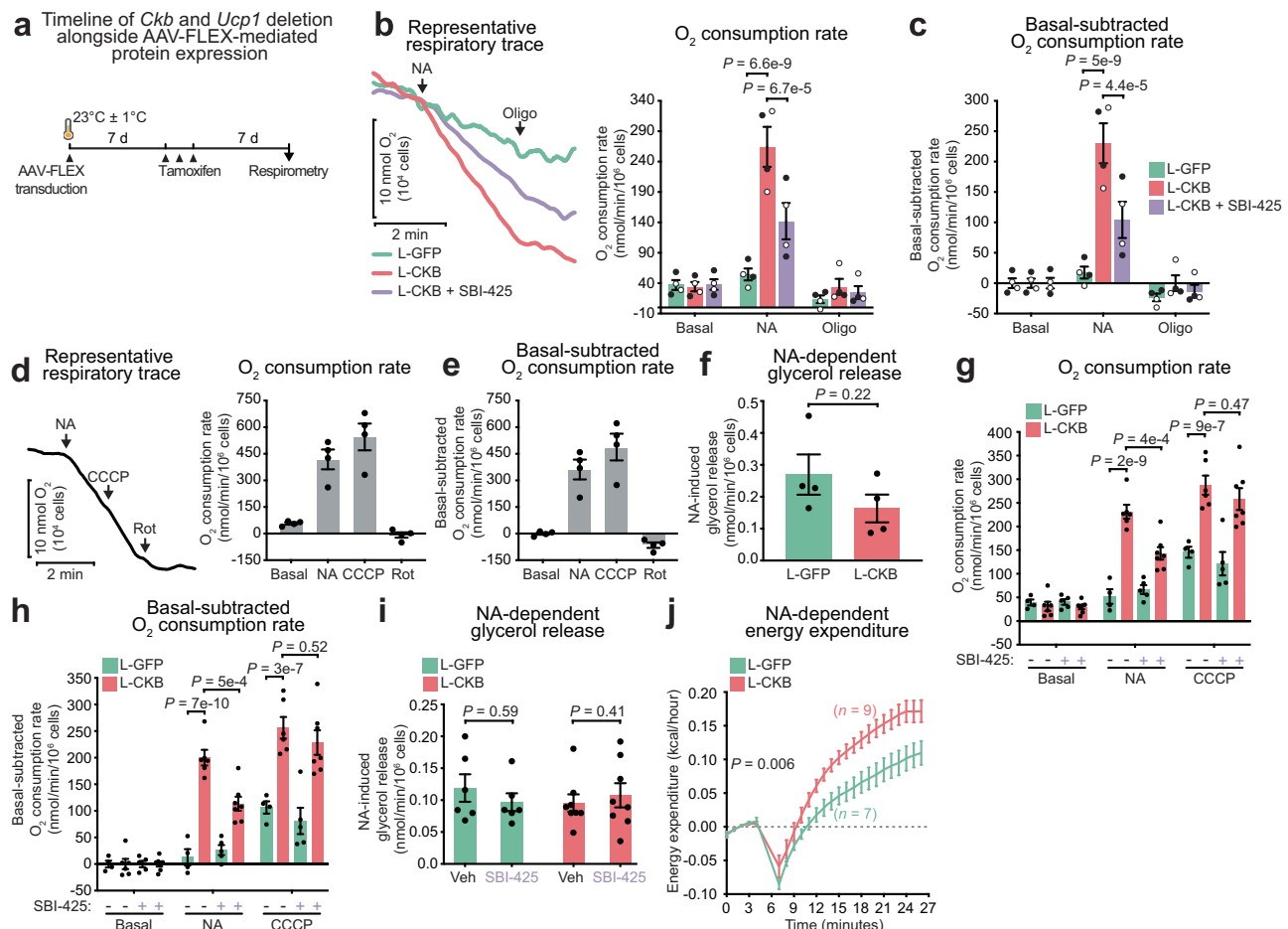

**Fig. 5 | Thermogenesis by the Futile Creatine Cycle in classical brown adipocytes. a** Schematic of AAV-FLEX transduction and tamoxifen-inducible protein expression in iADKO$^{Ckb;Ucp1}$ mice. Created in BioRender. Kazak, L. (2025) https://BioRender.com/j99d203. **b** Representative oxygen consumption traces of 10,000 brown adipocytes from 10- to 12-week-old iADKO$^{Ckb;Ucp1}$ mice in response to noradrenaline (NA, 0.1 μM), oligomycin (Oligo, 15 μM), and SBI-425 (10 μM, added at the start). Bar graphs show basal, followed by acute additions of NA and oligomycin ($n = 2$ independent preparations/sex). **c** Basal-subtracted oxygen consumption rates from Fig. 5b. **d** Representative oxygen consumption traces of 10,000 brown adipocytes from 16-week-old wild-type male mice in response to NA, CCCP (40 μM), and rotenone (Rot, 5 μM). Bar graphs show basal, followed by subsequent stimulation with NA, CCCP, and Rot ($n = 4$). **e** Basal-subtracted oxygen consumption rates from Fig. 5d. **f** NA-stimulated glycerol release from brown adipocytes from 12-week-old iADKO$^{Ckb;Ucp1}$ mice (NA, 10 min) ($n = 2$ independent preparations/sex/group). Data calculated from Supplementary Fig. 6a. **g** Oxygen consumption

rates of brown adipocytes from 16-week-old iADKO$^{Ckb;Ucp1}$ mice in response to NA and CCCP, grouped by treatments (L-GFP+veh, L-CKB+veh, L-GFP + SBI, L-CKB + SBI; $n = 4, 6, 5, 7$, respectively). **h** Basal-subtracted oxygen consumption rates from (**g**). **i** NA-stimulated glycerol release from brown adipocytes from 18-week-old iADKO$^{Ckb;Ucp1}$ mice (NA, SBI-425, 10 min) ($n = 6$: 2 females and 4 males for L-GFP groups; $n = 8$: 4 females and 4 males for L-CKB groups). Data calculated from Supplementary Fig. 6d. **j** in vivo NA-stimulated energy expenditure of 12-week-old iADKO$^{Ckb;Ucp1}$ mice expressing L-GFP or L-CKB in iBAT (females: $n = 4$ for L-GFP, $n = 4$ for L-CKB; males: $n = 3$ for L-GFP, $n = 5$ for L-CKB) acclimated to RT (23 °C ± 1 °C). **b**, **c**, **g**, **h** Two-way ANOVA (Tukey's post-hoc test); **f** two-tailed student's $t$ tests; **i** two-way ANOVA (Fisher's least significant difference (LSD)); **j** two-way ANOVA (Šidák post-hoc test). Data are presented as the mean ± s.e.m. and $n$ numbers are of biologically independent experiments. Source data are provided as a Source Data file.

similar impairment in cold-induced body temperature maintenance in mice lacking both *Ucp1* and *Ckb* in adipocytes[32].

## Thermogenesis by the FCC in classical brown adipocytes

Our findings thus far indicated that the FCC was sufficient to maintain whole-body homeothermy in response to cold. However, considering that cold responses are not confined to BAT in vivo, we sought to investigate the intrinsic thermogenic capacity of brown adipocytes mediated by the FCC using respirometry. Surgery-guided direct transduction of iBAT resulted in reduced variability of AAV-driven protein expression across different mice and a more uniform distribution compared to subcutaneous injection, due to the ability to directly visualize the iBAT during AAV delivery (Supplementary Fig. 5a-c). Although more invasive than subcutaneous injections, we prioritized this procedure because isolating mature brown adipocytes from iBAT requires pooling from multiple mice. Therefore, we

directly injected surgically exposed iBAT of room temperature (23 ± 1 °C)-acclimated iADKO$^{Ckb;Ucp1}$ mice of both sexes with AAV-FLEX-LACTB-GFP-FLAG or AAV-FLEX-LACTB-CKB-FLAG. One week after surgery, we induced L-GFP or L-CKB expression concurrently with endogenous *Ckb* and *Ucp1* co-deletion using tamoxifen. The following week, we examined the thermogenic capacity of acutely isolated brown adipocytes from these mice (Fig. 5a). Notably, stimulation with noradrenaline (NA) elicited an approximately 8-fold increase in the oxygen consumption rate of L-CKB-expressing iADKO$^{Ckb;Ucp1}$ brown adipocytes (Fig. 5b), resulting in a NA-dependent rate of 230.1 ± 32.9 nmol O$_2$/min$^{-1}$/10$^6$ cells, relative to basal respiration (Fig. 5c). In striking contrast, NA minimally induced respiration in L-GFP-expressing iADKO$^{Ckb;Ucp1}$ brown adipocytes (17.4 ± 9.8 nmol O$_2$/min$^{-1}$/10$^6$ cells, relative to basal respiration) (Fig. 5b, c). Therefore, the NA-stimulated respiratory response was approximately 13-fold higher in brown adipocytes expressing L-CKB compared to those

expressing L-GFP. Importantly, this thermogenic response was entirely dependent on mitochondrial ATP synthesis, as evidenced by its complete inhibition with acute oligomycin addition (Fig. 5b, c). Furthermore, TNAP inhibition significantly reduced NA-stimulated respiration in L-CKB-expressing brown adipocytes, dropping from $230.1 \pm 32.9$ nmol $O_2/min^{-1}/10^6$ cells to $104.2 \pm 29.9$ nmol $O_2/min^{-1}/10^6$ cells, in a manner dependent on ATP synthase (Fig. 5b, c). Therefore, TNAP-dependent thermogenesis accounted for 125.9 nmol $O_2/min^{-1}/10^6$ cells (Fig. 5c). These data provide additional evidence that the FCC substantially contributed to the respiratory response observed in L-CKB-expressing brown adipocytes. Unstimulated (basal) respiration was comparable between all groups (Fig. 5b). Next, we quantified NA-driven thermogenesis in wild-type brown adipocytes, expressing native UCP1 and CKB, to be $361.2 \pm 55.8$ nmol $O_2/min^{-1}/10^6$ cells, relative to basal respiration (Fig. 5d, e). These values are consistent with, or exceed, previously reported literature values[39,40]. This response was enhanced by 35% with the proton translocator CCCP (carbonyl cyanide 3-chlorophenylhydrazone), increasing to $488.2 \pm 75.4$ nmol $O_2/min^{-1}/10^6$ cells, relative to basal respiration (Fig. 5d, e), and was completely blocked by complex I inhibition with rotenone ($-65.2 \pm 15.3$ nmol $O_2/min^{-1}/10^6$ cells, relative to basal respiration) (Fig. 5d, e). Given that the TNAP-dependent thermogenic response to NA in L-CKB-expressing cells was 125.9 nmol $O_2/min^{-1}/10^6$ cells, and the full NA response in wild-type cells was $361.2 \pm 55.8$ nmol $O_2/min^{-1}/10^6$ cells, the FCC contributes a substantial portion of NA-mediated thermogenesis in these acutely isolated brown adipocytes. These results align within previous reports of the thermogenic contribution of TNAP and ATP turnover[28,40], via the FCC. These findings underscore that the cold tolerance conferred by L-CKB expression resulted from enhanced intrinsic thermogenesis in classical brown adipocytes. It was evident that L-CKB significantly enhanced the thermogenic capacity of brown adipocytes. However, genetic deletion of thermogenic effectors, like germline UCP1 deletion, while compromising thermogenesis directly also causes secondary changes in BAT that could impact overall thermogenic function. To determine whether restoring mitochondrial CKB not only reactivated the FCC but also broadly improved cellular responsiveness to NA, we first assessed NA-stimulated lipolysis. Brown adipocytes expressing L-GFP released more glycerol than those expressing L-CKB (Fig. 5f and Supplementary Fig. 6a), indicating that the enhanced thermogenesis in L-CKB cells was not driven by increased substrate mobilization. To further investigate, we repeated the brown adipocyte respiration experiments and tested the effect of CCCP, which provides an alternative route for proton re-entry into the mitochondrial matrix, bypassing specific thermogenic pathways. Interestingly, cells expressing L-CKB displayed a higher capacity for protonophore-mediated respiration ($256.2 \pm 20.3$ nmol $O_2/min^{-1}/10^6$ cells) compared to L-GFP-expressing cells ($106.3 \pm 11.5$ nmol $O_2/min^{-1}/10^6$ cells) (Fig. 5g, h). This finding suggests that the impaired maximal thermogenic response in cells lacking both UCP1 and CKB is partly rescued by restoring mitochondrial CKB. Indeed, a subunit of complex I (NDUFB8) and complex IV (MTCO1) showed significantly higher expression in purified brown adipocytes expressing L-CKB compared to those expressing L-GFP (Supplementary Fig. 6b, c). Again, pharmacological inhibition of TNAP significantly reduced NA-driven thermogenesis in L-CKB-expressing cells, while having no effect on L-GFP-expressing cells (Fig. 5g, h). Crucially, TNAP inhibition did not affect CCCP-mediated respiration (Fig. 5g, h) or NA-driven lipolysis in either L-GFP or L-CKB-expressing cells (Fig. 5i and Supplementary Fig. 6d), reinforcing that TNAP's role is specific to the FCC and its acute pharmacological inhibition does not influence mitochondrial respiratory capacity or substrate mobilization. Together, these results show that the enhanced thermogenic response in L-CKB-expressing cells stems from both a general increase in their capacity for substrate combustion and a heightened ability to drive

thermogenesis through the FCC. Finally, in vivo NA-induced thermogenesis revealed a significantly stronger response in iADKO[Ckb;Ucp1] mice expressing L-CKB compared to L-GFP within iBAT (Fig. 5j). Collectively, our data demonstrate that classical brown adipocytes possess a robust capacity for UCP1-independent thermogenesis, driven by substrate oxidation and ATP synthesis through the FCC and possibly FCC-independent pathways.

## Discussion

Most of our understanding of thermogenesis has been limited to studies on UCP1, constraining our knowledge of the quantitative contribution of UCP1-independent thermogenic pathways. Understanding these UCP1-independent pathways has been of long-standing interest[41–44] and is crucial for advancing our comprehension of BAT function and its broader implications for metabolic regulation and therapeutic strategies targeting metabolic disorders.

CKB, the principal creatine kinase isoenzyme in brown adipocytes[15,45], acts alongside UCP1 to regulate body temperature in the cold[32]. These findings highlight the key role of CKB in mouse thermogenic adipocytes, as no other creatine kinase isoenzyme compensates for its loss. However, prior to our work here, it was unclear whether CKB's impact on thermogenesis is derived from its role in the FCC or the Phosphocreatine Circuit. This distinction has implications for understanding its specific thermogenic function in brown adipocytes versus any secondary effects on adipocyte health. Furthermore, while experiments involving pan-adipose tissue loss-of-function indicated that CKB is necessary to maintain normal body temperature in the cold of mice lacking UCP1[32], the specific role of classical iBAT in this regulation remained ambiguous.

The prevailing view is that classical brown adipocytes do not perform thermogenesis to a meaningful level through an ATP-linked dissipative mechanism[46]. This tenet stems from a lower abundance of ATP synthase, relative to the electron transport chain, in BAT compared to other tissues like the liver and heart[18,47–49]. However, beyond ATP synthase expression levels in comparison to other tissues, this principle has not been empirically tested in any physiological context in vivo. Here, we have demonstrated that targeting CKB to mitochondria exclusively in parenchymal adipocytes within classical iBAT drives a sufficient rate of energy dissipation to compensate for the thermogenic impairments caused by the absence of native CKB and UCP1 in all adipose depots of the mouse. Since we exclusively rescued mitochondrial CKB, where it consumes rather than produces ATP, the thermogenic effects of CKB are likely mediated through the FCC. Consistent with this, pharmacological TNAP inhibition selectively reduced NA-driven respiration, a hallmark of thermogenic activation, while leaving CCCP-induced uncoupling, an artificial driver of respiration, and substrate availability unaffected. Furthermore, 75% of iADKO[Alpl;Ucp1] mice exhibited severe cold-induced hypothermia, closely matching cold-induced hypothermia observed in iADKO[Ckb;Ucp1] mice. In aggregate, these data underscore the importance of thermogenesis by the FCC in classical BAT.

One might anticipate that disrupting UCP1 and the FCC genetically would activate alternative futile cycles to compensate for their loss. However, our data show this is unlikely, as L-GFP-expressing iADKO[Ckb;Ucp1] brown adipocytes largely lacked NA-driven thermogenesis. Strikingly, mitochondrial CKB restoration was sufficient to drive an increase in maximal respiratory capacity in brown adipocytes. We hypothesize that this resulting increase in general mitochondrial function likely represents a secondary bioenergetic response to FCC-mediated thermogenesis, uncovering a component of respiration that is independent of both the FCC and UCP1. We define the FCC-independent component as the NA-mediated thermogenic response of mitochondrial CKB-expressing brown adipocytes that cannot be repressed by TNAP inhibition. Thus, our data in acutely isolated brown adipocytes indicate that UCP1-independent thermogenesis is mediated

substantially by the FCC and possibly by FCC-independent mechanisms. Three non-mutually exclusive possibilities that explain the FCC-independent component could be reflected by: 1) enhanced mitochondrial functionality, which would not be a dedicated energy-dissipative thermogenic pathway, 2) the activation of several minor pathways, or 3) the recruitment of a single alternative pathway. The precise nature of this FCC-independent pathway(s) await to be elucidated.

While UCP1 is primarily known for dissipating the proton motive force (PMF) to drive thermogenesis, this effect would undoubtedly influence other cellular functions. Mitochondrial proton leak alters the redox state of the respiratory chain and may lower reactive oxygen species (ROS) production[50,51]. In cold-acclimated brown fat, where respiratory chain complexes are abundant and ATP synthase is proposed to be limited, the absence of UCP1 is understood to prevent the release of excess PMF during thermogenic activation. We demonstrated that UCP1 inactivation in the germline causes substantial alterations in the BAT proteome, including dramatic reductions in respiratory chain proteins and adrenergic receptors, and leads to ROS-dependent disruptions in mitochondrial calcium buffering[32,52]. This aligns with reports showing that enhancing proton re-entry can mitigate ROS-induced mitochondrial dysfunction[51]. Our findings here suggest that mitochondrial CKB helps reduce these secondary changes in iADKO[Ckb;Ucp1] brown adipocytes, orthogonally supporting the idea of the FCC as a key thermogenic pathway. Since thermogenesis and mitochondrial capacity are positively correlated[32,39,51,52], in hindsight it is not surprising that mitochondrial CKB-restored iADKO[Ckb;Ucp1] brown adipocytes exhibit higher spare respiratory capacity than L-GFP-restored iADKO[Ckb;Ucp1] brown adipocytes. Overall, our findings indicate that UCP1-independent thermogenic pathways, such as the FCC, are sufficiently thermogenic to provide an alternative pathway for ATP synthase-mediated proton re-entry and sparing of mitochondrial respiratory capacity.

The iADKO[Alpl;Ucp1] mice display worsened cold intolerance compared to single *Ucp1* knockout mice, resembling the phenotype of iADKO[Ckb;Ucp1] mice, which is consistent with the involvement of the FCC. Notably, these data also suggest that FCC-independent pathways are insufficient to fully compensate in the absence of TNAP activity in vivo because most (75%) iADKO[Alpl;Ucp1] mice are cold-sensitive. However, if this sensitivity is caused by mitochondrial dysfunction rather than loss of the FCC, then it is very likely that the cold-sensitive phenotype of germline UCP1 knockout mice, which exhibit a more severe reduction in respiratory chain abundance[32], is primarily driven by mitochondrial dysfunction, not UCP1.

How would CKB and TNAP influence FCC-independent thermogenic pathways? The FCC differs from other futile cycles, such as calcium or lipid cycling, because it occurs solely within mitochondria[14,15,29,30], whereas those cycles involve non-mitochondrial compartments[53–56]. Cytosolic CKB could potentially support FCC-independent thermogenesis by maintaining high levels of ATP via the reverse creatine kinase reaction, but a functional mitochondrial creatine kinase that would generate the phosphocreatine in the first place would be required. However, our findings show that cytosolic CKB is not essential for thermogenesis, because mitochondrial CKB alone can restore thermogenesis in cells lacking both CKB and UCP1. Moreover, by catalyzing the synthesis of phosphocreatine, mitochondrial CKB would compete with alternative FCC-independent cycles for the available ATP pool. TNAP, known to hydrolyze phosphocreatine within mitochondria[14,15,28–30], also likely competes with FCC-independent thermogenesis by removing substrate required to drive the reverse creatine kinase reaction in the cytosol.

Collectively, our data reveal that classical brown adipocytes have the capacity to support physiologically significant thermogenesis via ATP turnover through the FCC-dependent pathway and through a FCC-independent pathway(s), challenging the conventional view that these cells rely solely on UCP1 for heat production.

## Methods

### Animals

Mouse experiments were performed according to procedures approved by the Animal Resource Centre at McGill University and complied with guidelines set by the Canadian Council of Animal Care. iADKO[Ckb;Ucp1] mice have been previously described[32]. *Ucp1*[fl/fl] (B6(129S4)-*Ucp1*[tm1c(EUCOMM)Hmgu]/KazlJ) and *Ckb*[fl/fl] (C57BL/6J-*Ckb*[em1Kazl]/J) mice will be available at the Jackson Laboratory Repository with the Jax Stock No. 039539 (https://www.jax.org/strain/039539) and 039540 (https://www.jax.org/strain/039540), respectively. AdipoqCre[+] and AdipoqCre[-] littermates were derived from breeding B6.FVB-Tg(Adipoq-cre)1Evdr/J (JAX stock No. 028020) with wild-type C57BL/6J (JAX stock No. 000664) mice. *Alpl*[fl/+] mice[29] were mated to *Ucp1*[fl/+] mice to generate *Alpl*[fl/+];*Ucp1*[fl/+] mice. *Alpl*[fl/+];*Ucp1*[fl/+] mice were then crossed to AdipoqCreERT2 mice[57] (C57BL/6 N, Jax Stock No. 025124) to generate: (1) inducible adipocyte-selective *Alpl;Ucp1* double knockout mice (iADKO[Alpl;Ucp1]), (2) control floxed mice (*Alpl*[fl/fl];*Ucp1*[fl/fl]), and (3) control AdipoqCreERT2 mice. Mice were maintained on a 12-hour light/dark cycle, with lights on from 07:00 to 19:00, with 07:00 defining Zeitgeber time 0 (ZT0). Animals had unrestricted access to drinking water and a standard chow diet (3.1 kcal/g energy density) composed of 24% protein, 16% fat, and 60% carbohydrate (2920X, Envigo, Madison, WI, USA). Mice were housed in groups of 3–5 per cage at 23 °C ± 1 °C with bedding and shredded paper strips provided until they reached the experimental stage (6–12 weeks of age). There is ongoing debate regarding the optimal housing temperature to best replicate human physiology in mice. Based on recommendations suggesting that providing bedding and nesting materials makes standard room temperature (22–24 °C) adequate[58], we maintained these conditions. For cold exposure experiments, mice were housed individually with bedding, maintaining ad libitum access to food and water. All experiments utilized age-matched littermates, with specific temperatures detailed in figure legends. Mice were euthanized via cervical dislocation, and tissues were either prepared for histological analysis or rapidly frozen in liquid nitrogen and stored at −80 °C for further study.

### Cloning

Flag-tagged *Ckb* (*Ckb-flag*) and *Gfp* (*Gfp-flag*) were cloned in the reverse orientation as SalI-HindIII fragments, after PCR amplification, into pAAV-CA-FLEX (Addgene, #38042) that had been linearized with HindIII and SalI. To clone *Lactb-Ckb-flag* and *Lactb-Gfp-flag*, the LACTB leader sequence was amplified by PCR from pAAV-LACTB-dAPEX2 (Addgene, #117179) as a SalI-BamHI fragment. *Ckb-flag* and *Gfp-flag* were amplified by PCR as BamHI-HindIII fragments. The *Lactb* insert, together with either the *Ckb-flag* or *Gfp-flag* insert, were cloned into pAAV-CA-FLEX that had been linearized with HindIII and SalI.

### Extraction of RNA

Frozen tissue samples were processed for total RNA extraction using TRIzol reagent (Ambion, Life Technologies, cat. No. 15596018), followed by purification with RNeasy Mini spin columns (Qiagen, cat. No. 74106), following the manufacturer's guidelines. The concentration and purity of the isolated RNA were assessed using a NanoDrop 8000 Spectrophotometer (Thermo Scientific Pierce, Waltham, Maine, USA).

### Reverse transcription-quantitative polymerase chain reaction (RT-qPCR)

RNA samples (1–2 µg) were converted to cDNA using the High-Capacity cDNA Reverse Transcription Kit (Applied Biosystems, cat. No. 4368814). The synthesized cDNA was subsequently analyzed by RT-qPCR. Each reaction contained 20 ng of cDNA along with 187.5 nmol of each primer, combined with GoTaq qPCR Master Mix (Promega, Cat. No. A6002). Reactions were performed in a 384-well plate format using the CFX384 Real-Time PCR System (Bio-Rad). Gene expression was

normalized via the ΔΔCt method, with *Ppib* mRNA serving as the reference gene. Data acquisition was carried out using CFX Maestro 2017 software. Primer sequences for RT-qPCR were as follows: *Ucp1* - Fwd: 5′ ACT GCC ACA CCT CCA GTC ATT 3′; Rev: 5′ CTT TGC CTC ACT CAG GAT TGG 3′; *Alpl* – Fwd: 5′ CCA ACT CTT TTG TGC CAG AGA 3′; Rev: 5′ GGC TAC ATT GGT GTT GAG CTT TT 3′; *Fabp4* – Fwd: 5′ AAG GTG AAG AGC ATC ATA ACC CT 3′; Rev: 5′ TCA CGC CTT TCA TAA CAC ATT CC 3′; *Ppib* – Fwd: 5′ GGA GAT GGC ACA GGA GGA A 3′; Rev: 5′ GCC CGT AGT GCT TCA GCT T 3′.

## AAV-FLEX Production
Adeno-associated virus (AAV) was produced using two 15 cm dishes of HEK293T/17 cells (ATCC, CRL-11268). Upon reaching 100% confluency, HEK293T/17 cells were transfected with 10 μg of pAAV-CA-FLEX (with the desired insertion), as well as 40 μg of packaging/helper plasmid, pDP8.ape (PlasmidFactory, PF0478), diluted in 2.5 ml of OptiMEM containing 200 μl of 1 mg ml⁻¹ polyethylenimine (Polyscience Inc., #23966-100) and the mixture was incubated for 10 min at room temperature before being added dropwise to the HEK293T/17 cells. Media was replaced 24 h following transfection. 72 h post-transfection, cells and media were harvested and centrifuged at 3000 $g$ for 15 min at 4 °C. The supernatant was filtered through a 0.45 μm PVDF filter and the virus was concentrated with 5x AAV Concentrator Reagent (System Biosciences, #AAV100A-1) for 24-48 h. The concentrated virus was pelleted at 1500 $g$ for 30 min at 4 °C and the pellet was resuspended in 0.4 ml sterile PBS, flash frozen in liquid nitrogen, and stored at −80 °C until further use.

## AAV-FLEX titering
AAV titer was assessed by qPCR[59]. In brief, pAAV-CA-FLEX was linearized using EcoRV (NEB, R3195), electrophoresed on a 0.5% agarose gel, purified using the QIAquick Gel Extraction Kit (Qiagen, 28704) and quantified using a NanoDrop 8000 Spectrophotometer (Thermo Scientific Pierce, Waltham, Maine, USA). Next, AAV DNA was purified from 5 μl of viral sample using DNA/RNA Extraction Reagent - ViralXpress™ (Millipore; 3095) following the manufacturer instructions. The DNA extracted from viral samples was quantified by qPCR against a standard curve of linearized pAAV-CA-FLEX. The primers used were: Forward, 5′- cgc tgc ttt aat gcc ttt gta t -3′; Reverse, 5′- ggg cca caa ctc ctc ata aa -3′. Primers were mixed with GoTaq qPCR Master Mix (Promega, A6002) and qPCR was performed in a 384-well format using a CFX384 Real-time PCR system (Bio-Rad).

## Preparation of avertin
To prepare the Avertin stock solution, 25 g of 2,2,2-tribromoethanol (Millipore Sigma, T48402) was dissolved in 15.5 ml of *tert*-amyl alcohol (Millipore Sigma, 152463). The mixture was heated to 50 °C until completely dissolved while being shielded from light. For the working solution (20 mg ml⁻¹), 0.5 ml of the stock solution was diluted in 39.5 ml of sterile isotonic saline. The solution was then heated to 40 °C until fully dissolved, filtered using a 0.2-μm filter.

## AAV injections subcutaneously above the interscapular BAT
Mice (genotypes, sex, and age detailed in figure legends) were anesthetized by an intraperitoneal injection of 0.5 g kg⁻¹ body weight Avertin (stock diluted in isotonic saline, injection volume: 25 μl g⁻¹ body weight). The interscapular region above the BAT was injected subcutaneously with a total volume of 200 μl containing $1 \times 10^{11}$–$1.5 \times 10^{11}$ viral infectious units.

## Inducible gene inactivation
Mice were housed at 23 ± 1 °C and maintained on a chow diet. At 6–9 weeks of age, three days following AAV-FLEX infection, tamoxifen (T5648, Sigma-Aldrich) was administered via intraperitoneal injection. The tamoxifen was prepared as a 20 mg.ml⁻¹ solution in corn oil and delivered at a dose of 75 mg kg⁻¹ once daily for three consecutive days at room temperature (23 ± 1 °C). After completing the treatment, mice were given a 4-day recovery period at the same temperature, totaling seven days post-injection. Subsequently, mice underwent acclimation to 30 °C for 7–9 days. During this period, they were transferred to metabolic cages set to 30 °C to adjust to the conditions and establish baseline metabolic parameters through indirect calorimetry. For other experimental protocols where metabolic cage measurements were not required, mice were handled identically, except that the 30 °C acclimation phase took place in rodent incubators (Powers Scientific), unless otherwise noted in the figure legends.

## Indirect calorimetry for cold challenge
After completing tamoxifen injections, mice remained at 23 ± 1 °C for a 4-day recovery period, making a total of seven days post-treatment. They were then gradually acclimated to 30 °C in rodent incubators for 3–6 days before being housed individually in metabolic cages (Promethion high-definition behavioral phenotyping system, Sable Systems International) under a 12-h light/dark cycle (lights on at 07:00, ZT0) for three additional days at 30 °C. To assess thermogenesis in response to cold exposure, baseline metabolic measurements were recorded over 24 h at 30 °C (with humidity maintained at 20–40%). The following morning (ZT0), the incubator temperature was abruptly lowered to 5 °C without a gradual transition, requiring ~75 min to reach the target temperature. Mice had continuous access to food and water throughout the experiment. Energy expenditure was recorded without adjusting for body weight using Sable Systems' data acquisition software (IM-3 v.23.0.4). Data analysis was performed with Macro-Interpreter software (v.23.6.0) and the 1-h One-Click Macro (v.2.51.0). Hypothermia was defined as a drop in energy expenditure below 0.2 kcal/h, corresponding to a rectal temperature of ~28 °C or lower, at which point ethical endpoints were met.

## In vivo noradrenaline-induced energy expenditure
Mice (with genotype, sex, and age specified in the figure legends) were anesthetized using an intraperitoneal injection of Avertin (0.5 g/kg body weight, diluted in isotonic saline, administered at 25 μl/g body weight). Anesthetized mice were then placed in individual metabolic cages (Promethion high-definition behavioral phenotyping system, Sable Systems International) set to 33 °C. Each experiment included data collection from four mice simultaneously. Following anesthesia, mice were positioned in the prone position within the metabolic cages. After an initial eight-minute baseline measurement of energy expenditure, the cages were briefly opened for subcutaneous administration of noradrenaline (Sigma-Aldrich, A9512) diluted in isotonic saline to a final concentration of 0.1 mg/ml. Each mouse received 200 μl of the solution, corresponding to a dose of 20 μg (60 nmol) noradrenaline, injected above the interscapular brown adipose tissue. Data analysis was performed using the 1-min One-Click Macro (v.2.51.0). Data was normalized to energy expenditure prior to NA injection.

## Western blot analysis
Tissue and cell samples were lysed in a buffer containing 50 mM Tris (pH 7.4), 500 mM NaCl, 1% NP40, 20% glycerol, 5 mM EDTA, and 1 mM PMSF, supplemented with Roche protease inhibitor cocktail. The homogenized samples were centrifuged at 16,000 $g$ for 10 min at 4 °C, and the resulting supernatants were collected for further analysis. Protein concentration was measured using the bicinchoninic acid (BCA) assay (Pierce, 23225). The optimal amount of protein loaded for each antibody was determined experimentally. Protein lysates were denatured in Laemmli buffer (60 mM Tris, pH 6.8, 2% SDS, 10% glycerol, 0.05% bromophenol blue, 0.7 M β-mercaptoethanol), resolved on 10% Tris/Glycine SDS-PAGE gels, and transferred onto polyvinylidene difluoride (PVDF) membranes. Primary antibodies were

diluted in a blocking solution consisting of TBS with 0.05% Tween-20 (TBS-T), 5% BSA, and 0.02% sodium azide, and membranes were incubated overnight at 4 °C. After washing, secondary antibodies were diluted in TBS-T containing 5% milk and applied for 45 min. Detection was performed using enhanced chemiluminescence (ECL) substrates (Bio-Rad), and signal visualization was carried out with a Bio-Rad Chemidoc Imaging System. The following antibody dilutions were used: VCL (Cell Signaling, clone E1E9V, 13901; 1:5000); UCP1 (Abcam, ab10983; 1:2000); CKB (Abcam, ab125114; 1:200; Abclonal, A12631; 1:1000); TNAP (R&D, AF2910; 1:200); TH (Millipore Sigma, ab152; 1:1000), total OXPHOS cocktail (Abcam, ab110413; 1:10,000), HSP60 (Abcam, ab46798; 1:10,000), LONP1 (Abcam; ab103809; 1:1500), Anti-rabbit (Promega, W4011; 1:10,000); Anti-mouse (Promega, W4021; 1:10,000). Band intensities were quantified using ImageJ software[60].

## Glycerol release assay

Freshly isolated brown adipocytes (10,000 cells in 0.3 ml) were resuspended in DMEM/F-12 medium (Thermo Fisher Scientific, 10565018) supplemented with 1% fatty acid-free BSA. To stimulate lipolysis, cells were incubated with 0.1 μM noradrenaline (NA) for 10 min at 37 °C. After incubation, glycerol released into the medium was separated from cells using a centrifugal filter (Millipore Sigma, UFC30LG25) by spinning at 8000 g for 30 s at room temperature. Glycerol concentration in the supernatant was quantified using the free glycerol reagent (Millipore Sigma, F6428) and a glycerol standard solution (Millipore Sigma, G7793), following the manufacturer's instructions.

## Histology

BAT was dissected and placed into tissue cassettes (Simport Scientific Inc, M505-2), followed by fixation in 10% zinc formalin (Fisher Scientific, 23-313095) at 4 °C overnight with gentle rocking. The next day, samples were washed three times in chilled PBS, then transferred to 70% ethanol and stored at room temperature (22 °C) in the dark to prevent light exposure. Tissue specimens were embedded in paraffin at the Rosalind & Morris Goodman Cancer Institute Histology Core Facility, McGill University. Paraffin-embedded blocks were sectioned into 5 μm thick slices and mounted onto glass slides. Formalin-fixed, paraffin-embedded (FFPE) slides were processed for deparaffinization and antigen retrieval using the Ventana Discovery Ultra system (Roche Diagnostics). Deparaffinization was carried out at 70 °C in EZ Prep solution (Roche, 950-102), while antigen retrieval was performed at 95 °C using cell conditioning 1 solution (Roche, 950-224).

## Sample preparation for epifluorescence microscopy

Sections were outlined with a PAP-pen and blocked with Sudan Black solution (0.3% Sudan black in 70% ethanol) for 20 min. Slides were then washed three times in PBS, 5 min each time, and blocked with Blocking reagent (Millipore Sigma, 20773) for 30 min in a humidified chamber at room temperature. Next, the slides were incubated with primary antibodies diluted in TBS containing 0.05% Tween (TBS-T), 1% BSA, overnight at 4 ° C in a humidified chamber, protected from light. Next, slides were washed three times with PBS and incubated for 30 min with secondary antibodies (diluted in TBS-T, 1% BSA) at room temperature in a humidified chamber, protected from light. Slides were washed three times with PBS and a coverslip (1.5 No. thickness) was mounted in Vectashield Plus Antifade Mounting Medium with DAPI (Vector Laboratories, H-2000) and sealed with transparent nail polish. Dilutions for antibodies were as follows: PLIN1 (Cell Signaling, 9349; dilution 1:200); FLAG M2 (Millipore Sigma, F1804; dilution 1:1000), Anti-rabbit AlexaFluor-488 (Thermo Fisher Scientific, A11008; dilution 1:200), Anti-mouse AlexaFluor-594 (Thermo Fisher Scientific, A11005; dilution 1:200).

## Widefield epifluorescence microscopy

Widefield images were acquired on a Zeiss Axio Observer inverted microscope running Zen Blue software, using a 20X 0.8 NA Plan-Apochromat air objective (Zeiss). Brightfield images were collected using transmitted light illumination with an exposure time of 10 ms, and fluorescence images were collected using an X-Cite 120 LED light source. DAPI, AlexaFluor-488, and AlexaFluor-594 were excited and images collected using FS 49, FS 10, and FS 15 filter sets (Zeiss) and exposure times of 200 ms, 150 ms, and 150 ms, respectively. Single-plane images for each channel were acquired on an Axiocam 506 m CCD camera (Zeiss) with 2 × 2 binning, with a resulting image pixel size of 0.454 μm.

## Confocal epifluorescence microscopy

Confocal images were collected on an inverted Stellaris-8 confocal microscope (Leica) running LAS X software, equipped with 100× 1.4 NA HC PL APO oil objective. The zoom was set to 0.75×, xy scanning dimensions were set to 2048 × 2048 pixels with scan speed at 400 Hz, resulting in a pixel size of 0.076 μm in xy, with a pixel dwell time of 0.7 μs, using 4x averaging. DAPI, AlexaFluor-488, and AlexaFluor-594 were excited, and images were collected using lasers at 405 nm (4% intensity, gain 12), 490 nm (2% intensity, gain 8), and 590 nm (2% intensity, gain 10) respectively. Brightfield pictures were taken using trans-PMT mode at gain set to 25.

## Sample preparation for STED super-resolution microscopy

Sections were outlined with PAP-pen. Slides were then washed three times in PBS, 5 min each time, and blocked with Blocking reagent (Millipore Sigma, 20773) for 30 min in a humidified chamber at room temperature. Next, the slides were incubated with primary antibodies diluted in TBS containing 0.05% Tween (TBS-T), 1% BSA, overnight at 4 °C in a humidified chamber, protected from light. Next, slides were washed three times with PBS and incubated for 30 min with secondary antibodies (diluted in TBS-T, 1% BSA) at room temperature in a humidified chamber, protected from light. Slides were washed three times with PBS and a coverslip (1.5 No. thickness) was mounted in Abberior Mount Solid Antifade (Aberrior, MM-2013), and sealed with transparent nail polish. Dilutions for antibodies were as follows: TOM20 (Santa Cruz, sc-11415; dilution 1:1000); FLAG M2 (Millipore Sigma, F1804; dilution 1:1000), Abberior STAR-RED, goat anti-rabbit (Abberior, STRED-1002; dilution 1:200), Anti-mouse AlexaFluor-594 (Thermo Fisher Scientific, A11005; dilution 1:200).

## STED super-resolution microscopy

Super-resolution images were collected on an inverted Stellaris-8 confocal microscope (Leica) running LAS X software, equipped with 100× 1.4 NA HC PL APO oil objective, a pulsed white light laser for excitation and a 775 nm depletion laser for STED. Briefly, for each image set, a confocal image and a STED image was acquired for each channel using line sequential scan mode. The zoom was set to 4×, xy scanning dimensions were set to 2048 × 2048 pixels, unidirectional scanning was used, scan speed was set to 400 Hz yielding a pixel dwell time of 0.7875 μs, and 6X line accumulation was used. The resulting images had a pixel size of 14.7 nm. AlexaFluor-594 was excited with a white light laser set to 594 nm with an intensity of 4%. Fluorescence emission was collected on a HyD X detector (in photon counting mode) with a range of 600 to 640 nm, with gating (using the TauSTED option) to keep photons with an arrival time of 1.7–2.9 ns. For the corresponding STED image, the 775 nm depletion laser was selected (in addition to the excitation laser) and set to 25%. STAR-RED was excited with a white light laser set to 640 nm with an intensity of 3%. Fluorescence emission was collected on a HyD S detector (in photon counting mode) with a range of 650–750 nm, with gating (using the TauSTED option) to keep photons with an arrival time of 0.8–4 ns. For

the corresponding STED image, the 775 nm depletion laser was selected (in addition to the excitation laser) and set to 30%.

## STED super-resolution microscopy quantification
The intensity measurements of the epifluorescence signal in Fig. 2e were carried out in Fiji[61] by applying the "ROI Manager - Measure" function to a line spanning across the ROI.

## FLAG$^+$ and PLIN1$^+$ quantitation
The percentages of cells that were positive or negative for FLAG and PLIN1 were counted manually using "Multi-point tool" in Fiji[61].

## Mitochondrial isolation from interscapular BAT
Mitochondria from BAT were isolated under cold conditions (4 °C, on ice). Interscapular BAT was collected from 5–8 mice, finely minced with scissors, and suspended in 10 ml of ice-cold SHE buffer containing 250 mM sucrose, 5 mM HEPES, and 1 mM EGTA, with the addition of 4% fatty acid-free BSA. The minced tissue was homogenized using a motorized Potter-Elvehjem Teflon pestle, and then passed through two layers of cheesecloth for filtration. The homogenate underwent centrifugation at $8500\,g$ for 10 min at 4 °C, after which the supernatant was quickly removed to discard the fat layer. The pellet was resuspended in 1 ml of ice-cold SHEB buffer before being diluted to a total volume of 20 ml with additional SHEB buffer. A low-speed centrifugation step was performed at $600\,g$ for 10 min at 4 °C to remove debris. The resulting supernatant was carefully transferred to a fresh tube and subjected to a second centrifugation at $8500\,g$ for 10 min at 4 °C to isolate mitochondria. The final mitochondrial pellet was resuspended in SHE buffer, and protein concentration was measured using the bicinchoninic acid (BCA) assay (Pierce). Prepared mitochondrial samples were then stored at −80 °C for future experiments.

## Creatine kinase activity
A coupled enzymatic reaction (pyruvate kinase and lactate dehydrogenase) was used to determine creatine kinase activity in the forward direction ($creatine + ATP \rightarrow ADP + phosphocreatine$). Absorbance at 340 nM was measured to determine the NADH oxidation rate using a BioTek Synergy H1 plate reader in kinetic mode. The assay was performed at 25 °C by supplementing assay buffer (20 mM $MgCl_2$, 100 mM KCl, 5 μM oligomycin (EMD Millipore, 495455), and 50 mM Tris, pH 9.0) with coupling substrates (5 mM ATP, 4 mM PEP, and 0.45 mM NADH), 40 μg of protein, and 10 mM creatine.

## Freshly isolated brown adipocytes
Interscapular BAT, was collected from mice (10-11 weeks old) that had been acclimated to room temperature (23 °C ± 1 °C). The tissue was finely minced and enzymatically digested in a modified Krebs-Ringer bicarbonate buffer (KRBMB) containing 135 mM NaCl, 5 mM KCl, 1 mM $CaCl_2$, 1 mM $MgCl_2$, 0.4 mM $K_2HPO_4$, 25 mM $NaHCO_3$, 20 mM HEPES, 10 mM glucose, and 4% fatty acid-free BSA. The digestion buffer was supplemented with collagenase B (2 mg/ml, Worthington) and soybean trypsin inhibitor (1 mg/ml, Worthington). Minced BAT from 4–5 mice was incubated in 10 ml of digestion buffer while continuously shaking at 37 °C for 50 min. The resulting cell suspension was passed through a 100-μm cell strainer to remove undigested tissue. To isolate mature adipocytes, the filtrate was allowed to float at room temperature for 10 min, followed by centrifugation at $80\,g$ for 5 min. Approximately 5 ml of the infranatant was withdrawn using a 10 ml syringe with an 18-gauge needle, and the stromal vascular fraction was discarded. The adipocytes were then washed three times with 5 ml of DMEM/F12 medium containing 10% FBS. After each wash, they were allowed to float for 20 min at room temperature before centrifugation at $80\,g$ for 5 min. After the final wash, tissue suspension was centrifuged at $80\,g$ for 5 min and the fat layer was immediately removed from the surface. The mature adipocytes, present below the fat layer, were allowed to float for 30 min before being collected from the surface of the infranatant, and moved into new tube. Cell number was determined using a Bright-Line Hemacytometer (Hausser Scientific).

## Respirometry of purified adipocytes
A Clark type electrode (Rank Brothers) was used to measure the oxygen consumption of adipocytes. DMEM/F12 supplied with 10% FBS was added to the chamber and left to equilibrate with atmospheric oxygen. Approximately 10,000 cells were then added to the chamber (0.7 ml final volume) maintained at 37 °C, covered with a lid and continuously stirred. The initial rate of cellular respiration prior to the addition of a thermogenic activator was termed "basal respiration". Drugs were added to the continuously stirring cells via a Hamilton syringe (0.1 μM noradrenaline (NA; Sigma, A9512), 15 μM oligomycin (Oligo95455)). To measure the acute effect of different drugs on respiration, the linear portion of the $O_2$ consumption rates were measured. The basal respiratory rate was subtracted from the drug-induced oxygen consumption rate to quantify NA-dependent and ATP-linked oxygen consumption rates. Multiple electrodes were used simultaneously to measure respiratory effects of distinct treatment groups in parallel, and different treatments were switched between electrodes to avoid any potential systematic bias coming from a single electrode. SBI-425 (MedChemExpress, HY-124756) was added at the beginning of the trace. Rank Brothers Dual Digital model 20: Picolog 6 data logging software was used for data collection.

## Surgical iBAT injection with AAV
Mice (genotypes, sex and age detailed in figure legends) were anesthetized using inhalant isoflurane (2.5% isoflurane at 0.8 l/min oxygen). Mice were injected with 500 μl of saline and carprofen (20 mg/kg body weight) prior to starting surgery. Additionally, the fur above the iBAT was shaved and disinfected with BAXEDIN (2% w/v chlorhexidine gluconate in 70% v/v isopropyl alcohol). When deep anesthesia was reached, a small (~0.5 cm) incision above the iBAT was made parallel to the spinal column. Next iBAT was separated from the subcutaneous surface by blunt dissection. iBAT was held in position and 4 injections (2.5 μl each) were performed on each lobe (total: 10 μl per lobe; 20 μl per mouse, with a total of $1 \times 10^{11}$–$1.5 \times 10^{11}$ viral infectious units). The incision site was treated with local analgesic (lidocaine) and sealed with surgical tissue glue (Vetbond). Mice were injected daily with carprofen for the next 2 days and their health was monitored afterwards for a duration of one week.

## Quantification of infection by surgical or subcutaneous AAV
Area-based quantification was carried out in Fiji[61] by first performing Otsu thresholding on the PLIN1 image to get a region of interest (ROI) of the entire area of the tissue, using the "Fill Holes" function. Then, to measure FLAG-positive areas, Otsu thresholding was performed on only the pixels in the FLAG image within the ROI that was generated with the PLIN1 image. The area of FLAG-positive cells were then divided by the total area of the tissue, and reported as a percentage.

## Reporting summary
Further information on research design is available in the Nature Portfolio Reporting Summary linked to this article.

# Data availability
All data supporting the findings described in this manuscript are available in the article and in the Supplementary Information and from the corresponding author upon request. Source data are provided with this paper.

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

## Acknowledgements

We thank all members of the Kazak laboratory for critical reading of the manuscript. This work was supported by Canadian Institutes of Health Research (CIHR) project grant (PJT-190219), a Natural Sciences and Engineering Research Council of Canada (NSERC) Discovery Grant, and the Canadian Foundation for Innovation John R. Evans Leaders Fund (37919) (to L.K.); a Michael D'Avirro Studentship and graduate scholarship from the Fonds de Recherche de Quebec – Santé (FRQS) (to J.B.); a NSERC Postgraduate Scholarship—Doctoral (PGS D) (to M.F.H.); a CIHR postdoctoral fellowship (MFE-176528) (to J.F.R.). L.K. is a Canada Research Chair in Adipocyte Biology. We acknowledge the histology core facility at the Rosalind & Morris Goodman Cancer Institute (GCI) at McGill University. Image data for this manuscript was collected at the Advanced BioImaging Facility (ABIF, RRID: SCR_017697) at McGill University, with the help of Joel Ryan.

## Author contributions

L.K. and J.B. conceived the project and designed the experiments. J.B., M.D-M., M.F.H., B.S., J.F.R., M.E., A.S., and L.K. performed the experiments. L.K. wrote the manuscript with editing contributions from all co-authors. L.K. supervised the project. These authors contributed equally: M.F.H. and M.D-M.

## Competing interests

The authors declare no competing interests.
