## [Transparent Peer Review file · Nature Communications]

The Futile Creatine Cycle powers UCP1-independent thermogenesis in classical BAT

Corresponding Author: Dr Lawrence Kazak

Version 0:

Reviewer comments:

Reviewer #2

(Remarks to the Author)

The authors reintroduce CKB into brown adipose tissue. Their findings show that CKB promotes thermogenesis via FCC. The authors provide a point-by-point response to previous reviews. Several points should be addressed before publication.

Major points:

-Reviewer#2 asked for a detailed analysis of the efficiency of subcutaneous vs direct intraparenchymal injection: Subcutaneous injection results in expression of the transgene only in the superficial area of BAT, whereas direct, "surgical" injection resulted in expression of the transgene throughout the tissue (ext data fig 6). A quantification (western blotting, imaging tools etc) is still missing and it seems (from the pictures shown) that there are fewer cells expressing the transgene after subcutaneous injection.

-The question about the specificity of this approach was also not adequately addressed: the authors still claim that "... FLAG expression was exclusively localized to perilipin 1-positive (PLIN1+) parenchymal brown adipocytes...". However, this conclusion is not substantiated by the data shown.

-Rev#2 asked about "oxygen consumption affected in BAT tissue samples of mice lacking UCP1 and CKB +/- CKB rescue" The authors try to address this question by studying adipocytes (not tissue) (fig 5). However, the graphs shown are not very informative and hard to decipher. It would be important to know the maximal respiration capacity of these cells upon FCCP treatment to demonstrate the cells are vital and comparable in terms of mitochondrial function. In addition, the authors could use antimycin A to block the mitochondria-related O₂ consumption. These drugs are used by many labs (classical biogenetic profiling) and would be informative (Oelkrug et al. 2013). Lastly, testing whether the NA stimulated O₂ consumption can be blocked by lipolysis inhibitor or not would be essential to exclude lipid futile cycling effects in this setup.

-Is the effect of CKB specific for BAT? To address this important question, the authors need to infect WATi and analyze the energy expenditure in vivo and ex vivo.
Not addressed.

-One major concern remaining is the bioenergetics analysis of whole animal brown fat function. Classically, this is done by measuring heat production upon NE or CL treatments. In addition, the authors used adiponectin Cre. Thus, it is hard to dissect whether the UCP1 KO cells or non-ucp1 expressing adipocytes are performing FCC thermogenesis.

Reviewer #3

(Remarks to the Author)

Version 1:

Reviewer comments:

Reviewer #2

(Remarks to the Author)

The authors performed new experiments, however there are still major issues :

-Energy expenditure:

I thank the authors for the efforts in revising the biogenetic analysis. Through the new experiments, the authors find that CKB expression enhanced mitochondria function during FCCP treatment (Fig. 5g, h). This indicates that CKB is essential for overall mitochondrial function.

It therefore can be envisioned that this impaired mitochondria function in the absence of CKB affected also other non-UCP1 dependent futile cycles. Thus, the conclusion that CKB/TNAP is the predominant futile cycle is not substantiated by the data shown.

This major issue is further underlined by the results shown in Fig. 5 j where they show that NA induced thermogenesis is still existing in CKB-deficient BAT (L-GFP), whereas CKB rescues or enhances only around 30% of energy expenditure.

-Analysis of the efficiency of subcutaneous vs direct intraparenchymal injection:

The authors now analyzed the efficiency of "surgical" vs "subcutaneous" injection: In the images shown, the "surgical" approach (max 83% transduced cells) appears to be superior to the subcutaneous approach (max 36%). However, a statistical analysis is missing. Are these representative images?

They also performed Western blotting, but in contrast to the imaging analyses, the differences appear to be small. This is puzzling.

Version 2:

Reviewer comments:

Reviewer #2

(Remarks to the Author)

The authors tried to discuss the 2 major points, but central questions are still open:

-Energy expenditure:

I thank the authors for the efforts to address this central point. However, the key issues are still not clarified: One key concern is that the loss of CKB impaired mitochondria function

This is shown in Fig 5B, where TNAP inhibition (via SBI-425) only partially blocked the effect of L-CKB overexpression (maximally 50%), even though the authors argue that the TNAP-specific effects achieved by SBI-425 is one of the key evidences to support their conclusion.

So taken together, the TNAP-dependent futile cycle is responsible for max. 50% of physiological, NA-induced energy expenditure (in the absence of UCP1). See also line 353 of the revised ms.

So the title and conclusion of the revised ms are still not correct and the TNAP-based futile cycle is not the "predominant UCP1-independent thermogenic pathway in classical BAT".

The authors also used a new mouse model to further consolidate their conclusion, however, here again, it cannot exclude the possibility that ALPL may impair mitochondria function in general and consequentially also other non-UCP1 dependent futile cycles.

Last but not least, the authors' interpretation of NE-response in UCP1 KO mice is also not convincing: The authors argue this is due not only to BAT but also to other sources. A different interpretation could be that there also other mechanism of UCP1-independent thermogenesis caused by several other futile cycles.

-Analysis of the efficiency of subcutaneous vs direct intraparenchymal injection:

The authors did not answer the central question: "how efficient is the new subcutaneous route of infection?" Now we have different numbers in the figure on page 49 for subq (23,23,36%) and 3 for iBAT injection (33,76, 83%) and two more numbers "approximately 64% efficiency compared to 27% for subcutaneous injections" are given. What kind of calculation was performed? How representative are these numbers and figures?

I specifically asked about the statistical analysis, which was also not answered. Again, what statistical analysis was performed on how many sections/images, are these differences significant? These are basic scientific standards.

Reviewer #3

(Remarks to the Author)

Based on the latest version of the paper, the claim that FCC is the predominant futile cycle is not supported by the data. This issue arises from the authors comparing two different mitochondrial types that have significantly different maximal capacities. Based on the data (5g), it appears that mitochondria lacking CKB have a maximal capacity that is only three times higher than the basal function, with only a slight increase observed following NA stimulation. Additionally, TNAP inhibition obstructs only 50% of this increase, which poses a concern given the high variation among the data points. Furthermore, the in vivo data (5j) does not align with the in vitro data.

Reviewer #2 (Remarks to the Author):

Major points:

1) R2 asked for a detailed analysis of the efficiency of subcutaneous vs direct intraparenchymal injection:

Subcutaneous injection results in expression of the transgene only in the superficial area of BAT, whereas direct, "surgical" injection resulted in expression of the transgene throughout the tissue (ext data fig 6). A quantification (western blotting, imaging tools etc) is still missing and it seems (from the pictures shown) that there are fewer cells expressing the transgene after subcutaneous injection.

RESPONSE TO REVIEWER:

Thank you for this comment, which led us to quantify our immunofluorescence images, now presented in a new **Extended Data Fig. 5**. We agree that the surgical method transduces a greater number of adipocytes. Because of this, we likely underestimated the thermogenic effects of the FCC in the cold challenge experiments, where we used subcutaneous injections. This highlights the importance of the FCC. In addition, we have now performed western blotting, comparing the expression of L-CKB following subcutaneous versus surgically-guided transduction into iBAT. This also shows that surgery increases the expression of AAV-encoded proteins. It is also important to note that this quantification has not affected any interpretation of our data.

2) The question about the specificity of this approach was also not adequately addressed:

The authors still claim that "...FLAG expression was exclusively localized to perilipin 1-positive (PLIN1+) parenchymal brown adipocytes...". However, this conclusion is not substantiated by the data shown.

RESPONSE TO REVIEWER:

Thank you for this insightful comment, which led us to quantify our immunofluorescence (IF) images. We measured the percentage of FLAG-positive cells among both PLIN1-positive and PLIN1-negative cells. Our new data reveal that 99% of FLAG-positive cells are also PLIN1-positive. These results are now presented in a new **Fig. 1 e, f** and new **Extended Data Fig. 1a**. In addition, a new **Fig. 1c**, shows biochemically that FLAG expression is exclusively localized to mature brown adipocytes without expression in the stromal vascular fraction. Together, these new data fully support our prior claim regarding the expected specificity of the Adiponectin-driven Cre model for targeting fat cells, consistent with its established role in the field.

3) R2 asked about "oxygen consumption affected in BAT tissue samples of mice lacking UCP1 and CKB +/- CKB rescue:

The authors try to address this question by studying adipocytes (not tissue) (fig 5). However, the graphs shown are not very informative and hard to decipher. It would be important to know the maximal respiration capacity of these cells upon FCCP treatment to demonstrate the cells are vital and comparable in terms of mitochondrial function. In addition, the authors could use antimycin A to block the mitochondria-related O₂ consumption. These drugs are used by many labs (classical biogenetic profiling) and would be informative (Oelkrug et al. 2013). Lastly,

testing whether the NA stimulated O₂ consumption can be blocked by lipolysis inhibitor or not would be essential to exclude lipid futile cycling effects in this setup.

RESPONSE TO REVIEWER:

Thank you for these insightful comments, which we have addressed with new experiments and reformatting of figures and figure legends.

3a. Bioenergetics using acutely isolated brown adipocytes, developed in the late 1960s, forms the foundation of our understanding of how brown adipocytes generate heat and has been used by the pioneering labs in our field: PMID: **4290444**; PMID: **5707712**; PMID: **5725810**; PMID: **9407062**; PMID: **6278947**; PMID: **176028**; PMID: **4314248**; PMID: **6295766**; PMID: **10825155**; PMID: **549646**. The bioenergetic profiling of isolated brown adipocytes is conducted immediately after their extraction from the tissue. The graphs we present follow standard bioenergetics protocols, typically displaying a representative trace alongside bar graphs (see all references above). Numerous studies, including those from our own lab, have utilized this approach without causing confusion for reviewers or investigators. All necessary details are provided in the figure legends and our thoroughly detailed methods section.

3b. Some investigators measure iBAT temperature, but this is a poor choice because it is neither quantitative nor accurate, as temperature can be altered by non-thermogenic means, such as changes in thermal insulation and blood flow (discussed in detail in these excellent reviews: PMID: **23760815**; PMID: **21177944**). In addition, isolated cells, as we have used, attain a much higher degree of thermogenic stimulation than tissue fragments/slices (PMID: **5725810**). In our data, we detected a 13-fold increase in respiration in response to noradrenaline in L-CKB expressing iADKO brown adipocytes.

3c. To assess the effect of mitochondrial CKB on BAT thermogenesis at the tissue level, we examined noradrenaline-induced thermogenesis *in vivo*. Noradrenaline triggered a significantly stronger thermogenic response in L-CKB-expressing iADKO BAT compared to BAT expressing L-GFP (new **Fig. 5j**). The difference in response resulted in a similar magnitude to what we have shown for UCP1 (PMID: **38272036**).

3d. We also want to thank the reviewer for prompting us to explore the possibility of general adipocyte functions. We have now expanded our analyses to study lipolysis, which is also, in addition to mitochondrial respiratory capacity, a key intermediate within the thermogenic response. The iADKO brown adipocytes restored with L-GFP show higher NA-dependent glycerol release than those restored with L-CKB. This finding reinforces the significance of the FCC, because even though there was less substrate mobilization, they were more thermogenic than L-GFP-restored cells.

We assessed mitochondrial respiratory chain abundance and maximal respiratory capacity in iADKO mice expressing L-GFP or L-CKB using western blotting for electron transport chain proteins (new **Extended Data Fig. 6b, c**) and chemical uncouplers, respectively (new **Fig. 5 g, h**). Interestingly, a subunit of complex IV was elevated in L-CKB-expressing brown adipocytes compared to L-GFP-expressing adipocytes. In addition, we show that L-CKB-expressing cells have a greater capacity for chemical uncoupling-mediated respiration (new **Fig. 5 g, h**). Bottom line is that the enhanced thermogenic response in L-CKB-expressing cells results from both an

increased capacity for substrate combustion and an improved ability to drive thermogenesis through the FCC—likely with the latter driving the former. The interpretation of these new data are discussed in the manuscript.

3e. Rotenone, a complex I inhibitor, fully blocks the NA and CCCP response (**new Fig. 5d, e**), consistent with NADH-driven respiration that would occur through fatty acid oxidation.

3f. Lipolysis and subsequent fatty acid oxidation fuel the citric acid cycle, proton pumping, and mitochondrial oxygen consumption. Therefore, blocking lipolysis is not a diagnostic to test the Futile Lipid Cycle, or any thermogenic pathway specifically, as fat breakdown is essential for all thermogenic pathways, the most famous one being UCP1. Oxygen consumption and fatty acid levels track immaculately well, both increasing during thermogenesis and both decreasing when thermogenesis is terminated (PMID: **6273408**). Moreover, the addition of fatty acids to brown adipocytes mimics the NA response (PMID: **6273408**). Effective activation of any UCP1-independent thermogenic pathway requires blocking the ATP-consuming step. This is why inhibiting TNAP gives us confidence that the increased thermogenesis in L-CKB-expressing cells is specifically through the FCC. As detailed below, TNAP inhibition had no impact on lipolysis or respiration driven by CCCP. These findings support that TNAP controls the FCC, rather than other pathways that can affect respiration.

If restoring mitochondrial CKB specifically activated the Futile Lipid Cycle (which involves both mitochondrial and cytosolic processes), TNAP would also need to be involved, since SBI-425 blocked a significant portion of CKB-driven thermogenesis (as shown in **Fig. 5b, c** of our prior submission and in a **new Fig. 5g, h**). Specifically, we tested whether SBI-425 (a selective TNAP inhibitor) affects lipolysis. However, SBI-425 did not impair lipolysis in either L-GFP- or L-CKB-expressing brown adipocytes (**new Fig. 5i** and **new Extended Data Fig. 6d**). Since inhibiting TNAP pharmacologically reduces thermogenesis in CKB-expressing cells without affecting lipolysis, this disproves the hypothesis that L-CKB restoration triggers futile lipid cycling. Additionally, brown adipocytes that express L-CKB exhibit reduced NA-mediated lipolysis, compared to L-GFP-expressing cells, yet despite this acquire an enhanced capacity for thermogenesis, further refuting an involvement of a futile lipid cycle.

3g. Loss of UCP1 and CKB causes brown fat cells to have zero response to noradrenaline. Thus, there is no thermogenic pathway occurring in these cells. When we restored only mitochondrial CKB, we detected robust thermogenesis. Therefore, if this thermogenic response was due to the Futile Lipid Cycle, it should be dependent on CKB. However, mitochondrial CKB, as utilized in our study, does not generate ATP but consumes it to produce PCr (**Referee-only Figure 1**). So, this is opposite of what

would be expected if it were fueling the Futile Lipid Cycle. Similarly, the hydrolysis of PCr by TNAP would support the CKB reaction, effectively also attacking the ATP pool. Therefore, based on the

Referee-only Figure 1. This figure depicts the Futile Creatine Cycle (FCC) and the Futile Lipid Cycle (FLC). The FLC is proposed to be thermogenic because ATP fuels the re-esterification arm of this cycle. The FCC, similarly uses ATP to dissipate energy. CKB is the ATP consuming step, which is linked to TNAP that hydrolyzes PCr.

well-established enzymatic properties of CKB and TNAP, TNAP inhibition should increase the

availability of ATP, which it does (PMID: 33981039), which would drive thermogenesis via the Futile Lipid Cycle – if it exists in brown adipocytes. However, the opposite is true, TNAP inhibition impairs thermogenesis, so this is not consistent with a model where L-CKB somehow promotes Futile Lipid Cycling. If, in addition to PCr, ATP was also a TNAP substrate, again, this reaction would compete with the Futile Lipid Cycle, or any ATP-consuming thermogenic pathways that is independent of the FCC. Together, there is no evidence that lipid cycling is occurring in our cells, and in fact our data demonstrate that it is unlikely.

4) Is the effect of CKB specific for BAT?

To address this important question, the authors need to infect WATi and analyze the energy expenditure *in vivo* and *ex vivo*. Not addressed.

RESPONSE TO REVIEWER: Thank you for your comment, and we apologize for any confusion. Our study is specifically focused on BAT and its role in supporting ATP-coupled thermogenesis via the FCC. We are not comparing BAT to WAT, as that would be a separate project. While WAT is of interest to us, it falls outside the scope of this manuscript and does not impact the interpretations we have made in this work. To address whether WAT impacts thermogenesis, this is not trivial, as there are multiple WAT depots (PMID: 25898951), such as: interscapular WAT, anterior subcutaneous WAT, triceps-associated WAT, inguinal WAT, mesenteric WAT, epididymal WAT, perirenal WAT, cardiac WAT (Referee-only Figure 2).

If we need to assess all WAT depots, we would require intravenous AAVs along with BAT removal to rule out BAT interference. Selective targeting of WAT is feasible with only one or possibly two depots, but attempting to address all depots simultaneously becomes increasingly complex.

Referee-only Figure 2. This figure depicts all the fat depots in a mouse, taken from PMID: 25898951.

5) One major concern remaining is the bioenergetics analysis of whole animal brown fat function.

Classically, this is done by measuring heat production upon NE or CL treatments.

RESPONSE TO REVIEWER: Thank you for this fantastic suggestion, which prompted us to perform NA injections *in vivo*. Our data (**new Fig. 5j**) now demonstrate that L-CKB restoration in iADKO BAT boosts NA-driven thermogenesis to a significantly greater extent than L-GFP restoration, and to a similar magnitude to what we have shown for UCP1 (PMID: 38272036).

6) In addition, the authors used adiponectin Cre. Thus, it is hard to dissect whether the UCP1 KO cells or non-ucp1-expressing adipocytes are performing FCC thermogenesis.

RESPONSE TO REVIEWER: We had not previously considered the possibility of adipocytes in classical BAT that do not express UCP1. This comment led us to investigate this further. After analyzing single-cell data, we found minimal evidence of non-UCP1-expressing classical brown adipocytes. Specifically, 90-95% of Adipoq+ cells are also Ucp1+ (PMID: **33846638**; PMID: **31573981**). While there may be a very minor population of Adipoq+ cells that apparently lack UCP1, this population probably does express UCP1, just at levels below the detection limit. In any case, this population, if it exists, is minor. Given the substantial thermogenesis we observe both in cells and *in vivo*, and the fact that our immunofluorescence images show that more than 5% of adipocytes were infected, it is likely that our results pertain primarily to adipocytes that naturally express UCP1, which is the more relevant population.

We thank the reviewer for their feedback and for raising points that allow us to clarify our findings further.

Reviewer #2 (Remarks to the Author):

Major points:

1) *Energy expenditure:*

I thank the authors for the efforts in revising the biogenetic analysis. Through the new experiments, the authors find that CKB expression enhanced mitochondria function during FCCP treatment (Fig. 5g, h). This indicates that CKB is essential for overall mitochondrial function. It therefore can be envisioned that this impaired mitochondria function in the absence of CKB affected also other non-UCP1 dependent futile cycles. Thus, the conclusion that CKB/TNAP is the predominant futile cycle is not substantiated by the data shown. This major issue is further underlined by the results shown in Fig. 5 j where they show that NA induced thermogenesis is still existing in CKB-deficient BAT (L-GFP), whereas CKB rescues or enhances only around 30% of energy expenditure.

RESPONSE TO REVIEWER: Below, we address the key critiques, integrating additional experimental evidence and detailed interpretations to support our conclusions.

1. Evidence supporting FCC as a Predominant Thermogenic Pathway.

In our cellular bioenergetic assays, we define the FCC as the NA-mediated thermogenic response of mitochondrial CKB-expressing brown adipocytes that is repressed by the highly specific TNAP inhibitor SBI-425.

- *TNAP-specific Effects:* TNAP inhibition (via SBI-425) significantly reduces NA-stimulated oxygen consumption in mitochondrial CKB-expressing cells by approximately **50%** (from 200–230 nmol O₂/min/million cells to ~104–113 nmol O₂/min/million cells), while having essentially no effect in GFP-expressing cells (**Fig. 5c, h**). Notably, TNAP inhibition selectively reduces NA-driven respiration—a hallmark of thermogenic activation—while leaving CCCP-induced uncoupling (**Fig. 5h**), an artificial driver of respiration, and substrate liberation (**Fig. 5i**), as measured by lipolysis, unaffected. This calculated contribution and specificity for NA-mediated thermogenesis by TNAP provides the evidence by which we make our interpretation: that the FCC is a major thermogenic contributor.
- *Cold Intolerance in a new FCC-Deficient Model:* We have collected new data showing that mice with co-deletion of *Alpl* and *Ucp1* (iADKO^{Alpl;Ucp1}), exhibit severe cold intolerance, with **75%** of these mice becoming hypothermic (**Fig. 4m, n**). This phenotype exceeds that observed in mice with *Ucp1* deletion alone¹. The findings that co-deleting *Ckb* or *Alpl* with *Ucp1* produces similar hypothermic phenotypes is consistent with the effect being due to the FCC. Notably, these data also suggest that FCC-independent pathways are insufficient to fully compensate in the absence of TNAP activity *in vivo* because most iADKO^{Alpl;Ucp1} mice, 75%, are cold-sensitive, further establishing the FCC as the predominant UCP1-independent thermogenic pathway.

2. FCC-independent Thermogenesis and Mitochondrial Function.

We agree with the reviewer that an FCC-independent NA response exists only when mitochondrial CKB is restored. **Notably, GFP-expressing cells show essentially no thermogenic response, suggesting that these alternative cycles are not recruited when CKB is absent.** The restoration of mitochondrial CKB reveals thermogenesis through both the FCC (the TNAP-inhibitable portion of NA-mediated thermogenesis in mitochondrial CKB-expressing cells) and FCC-independent pathways. As we describe below, the ability of mitochondrial CKB to enhance mitochondrial respiratory capacity does not refute the impact of the FCC, but in fact provides orthogonal support of it, as it demonstrates an ability–like UCP1²–of the FCC to regulate overall mitochondrial function.

As we note in our paper, we acknowledge the elevated maximal respiratory rates in mitochondrial CKB-expressing cells during CCCP treatment. Mitochondrial disruptions are established to occur in thermogenically-impaired BAT, such as in germline UCP1-deficient models^{2,3}. Given the connection between thermogenic pathways and mitochondrial function, with ROS as an intermediate^{2,4}, it is not surprising that restoring mitochondrial CKB in cells lacking native *Ckb* and *Ucp1*, but with endogenous TNAP, could maintain mitochondrial health by reactivating the FCC. **In fact, we interpret these data as strong orthogonal evidence supporting the significance of the FCC.** We speculate that proton re-entry by thermogenic pathways relieves the thermodynamic backpressure resulting in reduced ROS-induced damage. This is well worth pursuing in future projects.

We define the FCC-independent response in cellular bioenergetic experiments as the NA-mediated thermogenic response of mitochondrial CKB-expressing brown adipocytes that cannot be repressed by TNAP inhibition. Three non-mutually exclusive possibilities explain the FCC-independent component:

2.1 Enhanced Mitochondrial Function: The FCC-independent component of respiration might stem from improved mitochondrial functionality, which would not be a dedicated energy-dissipative thermogenic pathway. In this case, our interpretation that the FCC is the predominant thermogenic pathway remains valid.

2.2 Multiple Minor Pathways: The FCC-independent portion of respiration may reflect contributions from several minor thermogenic pathways. c

2.3 Single Pathway: The FCC-independent response might result from a single pathway contributing equally to thermogenesis as the FCC. As shown in our data (**Fig. 5c, h**), the FCC accounts for 50% of the UCP1-independent thermogenic response. Therefore, this hypothetical single pathway could, at most, contribute the remaining 50%, making its contribution equal to that of the FCC. If this scenario were true, it would necessitate revising our title to describe the FCC as *a major* UCP1-independent pathway rather than the predominant one. However, we believe that alternative explanations (as outlined in sections 2.1 and 2.2) and our *in vivo* iADKO^{Alpl;Ucp1} data argue against this possibility, making it less likely.

3. NA-Induced Thermogenesis in CKB-deficient BAT *in vivo*.

The 36% difference between GFP and mitochondrial CKB *in vivo* (**Fig. 5j**) aligns with findings from previous studies (**Reviewer-only Fig. 1b–e**)^{1,5-8} and is consistent with the expected effects of noradrenaline (NA), which stimulates energy expenditure from multiple sources, not just BAT. Notably, whole-body oxygen consumption increases with NA treatment even when interscapular

BAT is completely removed (**Reviewer-only Fig. 1f**)⁹, highlighting NA's role in activating thermogenesis in non-brown adipocytes. Consequently, the observed thermogenic effects cannot be fully attributed to BAT. Thus, a 36% reduction, as observed here, is substantial and comparable to other significant genetic perturbations (**Reviewer-only Fig. 1b–e**).

Reviewer only Fig 1. a, Noradrenaline-dependent thermogenic response of BAT and non-BAT. Data taken from Fig. 1 of Hirata, *J J Physiol*, 1982. **b**, Noradrenaline-dependent thermogenic response between wild type, Ucp1(+/+), and Ucp1 knockout, Ucp1(-/-) mice. Data taken from Fig. 2c of Feldmann et al, *Cell Metabolism*, 2009. **c**, Noradrenaline-dependent thermogenic response between wild type (Ucp1^{+/+}, and Ucp1^{-/-} mice. Data taken from Fig. 1p and Fig. 1q of Rahbani and Bunk et al, *Cell Metabolism*, 2024. **d**, Noradrenaline-dependent thermogenic response between wild type (WT), Ucp1 knock-in heterozygotes (Ucp1^{KI/+}) and Ucp1 knock-in homozygotes (Ucp1^{KI/KI}) mice. Data taken from Fig. 4b, d of Wang et al, *AJP Endo & Metab*, 2021. **e**, Noradrenaline-dependent thermogenic response between wild type (WT) and fat-specific Ebf2 knockout (Ebf2^{DA^{dip}oq}) mice. Data taken from Fig. 1g of Angueira et al, *Cell Reports*, 2020. **f**, The data shows the thermogenic response (ml O₂/100cm² in 30 min) of rats that have interscapular BAT (sham) or rats where the interscapular BAT was removed (-IBAT). Data taken from Fig. 2 of Himms-Hagen et al, *Adv Enzyme Regul*, 1970.

2) Analysis of the efficiency of subcutaneous vs direct intraparenchymal injection:

The authors now analyzed the efficiency of “surgical” vs “subcutaneous” injection: In the images shown, the “surgical” approach (max 83% transduced cells) appears to be superior to the subcutaneous approach (max 36%). However, a statistical analysis is missing. Are these representative images? They also performed Western blotting, but in contrast to the imaging analyses, the differences appear to be small. This is puzzling.

RESPONSE TO REVIEWER: We appreciate the reviewer's comments and the opportunity to clarify.

Western blotting is inherently semi-quantitative and does not account for protein expression levels specifically within transduced cells. Therefore, it is not expected that western blot results will perfectly match transduction efficiency percentages, even though they show relatively close agreement. Importantly, this discrepancy does not affect the interpretation of our data. Instead, it highlights that the surgical method achieves a higher degree of transduction efficiency in the tissue. Whether the two metrics align is irrelevant to the conclusions drawn.

When averaging transduction efficiencies across dorsal-ventral BAT sections, surgical injections achieve approximately 64% efficiency compared to 27% for subcutaneous injections, making the surgical method about 2.4 times more effective. Meanwhile, western blot analysis of whole BAT lobe lysates shows a ~1.5-fold increase in AAV-mediated expression with surgical injections. These results are broadly consistent and reinforce the conclusion that the surgical approach is superior.

CONCLUDING REMARKS: To refine our interpretation and address the reviewer's concerns, we have clarified in the manuscript that while the FCC represents a major contributor to UCP1-independent thermogenesis, our data do not exclude the existence or relevance of parallel non-UCP1 mechanisms. We maintain that our conclusions of the key role of the FCC remain well-supported by the data and reflect the multifaceted nature of thermogenic regulation. We thank the reviewer again for their constructive feedback.

References

- 1 Rahbani, J. F. *et al.* Parallel control of cold-triggered adipocyte thermogenesis by UCP1 and CKB. *Cell Metab*, doi:10.1016/j.cmet.2024.01.001 (2024).
- 2 Matthias, A. *et al.* Thermogenic responses in brown fat cells are fully UCP1-dependent. UCP2 or UCP3 do not substitute for UCP1 in adrenergically or fatty acid-induced thermogenesis. *J Biol Chem* **275**, 25073-25081, doi:10.1074/jbc.M000547200 (2000).
- 3 Kazak, L. *et al.* UCP1 deficiency causes brown fat respiratory chain depletion and sensitizes mitochondria to calcium overload-induced dysfunction. *Proc Natl Acad Sci U S A* **114**, 7981-7986, doi:10.1073/pnas.1705406114 (2017).
- 4 Oelkrug, R., Kutschke, M., Meyer, C. W., Heldmaier, G. & Jastroch, M. Uncoupling protein 1 decreases superoxide production in brown adipose tissue mitochondria. *J Biol Chem* **285**, 21961-21968, doi:10.1074/jbc.M110.122861 (2010).
- 5 Hirata, K. Enhanced calorogenesis in brown adipose tissue in physically trained rats. *Jpn J Physiol* **32**, 647-653, doi:10.2170/jjphysiol.32.647 (1982).
- 6 Feldmann, H. M., Golozoubova, V., Cannon, B. & Nedergaard, J. UCP1 ablation induces obesity and abolishes diet-induced thermogenesis in mice exempt from thermal stress by living at thermoneutrality. *Cell Metab* **9**, 203-209, doi:10.1016/j.cmet.2008.12.014 (2009).
- 7 Wang, H. *et al.* Uncoupling protein-1 expression does not protect mice from diet-induced obesity. *Am J Physiol Endocrinol Metab* **320**, E333-E345, doi:10.1152/ajpendo.00285.2020 (2021).

- 8 Angueira, A. R. *et al.* Early B Cell Factor Activity Controls Developmental and Adaptive Thermogenic Gene Programming in Adipocytes. *Cell Rep* **30**, 2869-2878 e2864, doi:10.1016/j.celrep.2020.02.023 (2020).
- 9 Himms-Hagen, J. Regulation of metabolic processes in brown adipose tissue in relation to nonshivering thermogenesis. *Adv Enzyme Regul* **8**, 131-151, doi:10.1016/0065-2571(70)90013-0 (1970).

Reviewer #2 (Remarks to the Author):

2.1. I thank the authors for the efforts to address this central point. However, the key issues are still not clarified: One key concern is that the loss of CKB impaired mitochondria function This is shown in Fig 5B, where TNAP inhibition (via SBI-425) only partially blocked the effect of L-CKB overexpression (maximally 50%), even though the authors argue that the TNAP-specific effects achieved by SBI-425 is one of the key evidences to support their conclusion. So taken together, the TNAP-dependent futile cycle is responsible for max. 50% of physiological, NA-induced energy expenditure (in the absence of UCP1). See also line 353 of the revised ms. So the title and conclusion of the revised ms are still not correct and the TNAP-based futile cycle is not the “predominant UCP1-independent thermogenic pathway in classical BAT”.

The authors also used a new mouse model to further consolidate their conclusion, however, here again, it cannot exclude the possibility that ALPL may impair mitochondria function in general and consequentially also other non-UCP1 dependent futile cycles.

RESPONSE TO REVIEWER: We are in agreement that the data presented in our paper show that 50% of UCP1-independent thermogenesis in our model is controlled by the FCC and the remainder is mediated by one or more FCC/UCP1-independent pathways. We also agree that while co-deletion of TNAP and UCP1 impairs thermogenesis through the FCC, it is possible that general mitochondrial dysfunction contributes to the thermogenic impairment of iADKO^{Alpl/Ucp1} mice. All mentions of the FCC being “predominant” have been removed.

2.2. Last but not least, the authors’ interpretation of NE-response in UCP1 KO mice is also not convincing: The authors argue this is due not only to BAT but also to other sources. A different interpretation could be that there also other mechanism of UCP1-independent thermogenesis caused by several other futile cycles.

RESPONSE TO REVIEWER: Agreed, the NA response in UCP1 KO is not the relevant data that addresses this point. This was demonstrated in our prior response where we cited and included a figure from a key study that showed an increase in whole-body oxygen consumption with NA treatment even when interscapular BAT was completely removed. We include this figure again for the reviewer and editor (**Reviewer only Fig. 1**)¹. While we agree with the reviewer that there are alternative mechanisms for UCP1-independent thermogenesis driven by other futile cycles, we presented the UCP1 KO data to address the reviewer’s claim that the effect size we observed in vivo was small. The data from the UCP1 KO mice clearly show that our effect size is on par with published literature of established thermogenic effectors (**Reviewer only Fig. 2**).

Reviewer only Fig 1. The data shows the thermogenic response (ml O₂/100cm² in 30 min) of rats that have interscapular BAT (sham) or rats where the interscapular BAT was removed (-IBAT). Data taken from Fig. 2 of Himms-Hagen et al, *Adv Enzyme Regul*, 1970.

2.3. Analysis of the efficiency of subcutaneous vs direct intraparenchymal injection: The authors did not answer the central question: “how efficient is the new subcutaneous route of infection?” Now we have different numbers in the figure on page 49 for subq (23,23,36%) and 3 for iBAT injection (33,76, 83%) and two more numbers “approximately 64% efficiency compared to 27% for subcutaneous injections” are given. What kind of calculation was performed? How representative are these numbers and figures? I specifically asked about the statistical analysis, which was also not answered. Again, what statistical analysis was performed on how many sections/images, are these differences significant? These are basic scientific standards.

RESPONSE TO REVIEWER: Absolutely, scientific standards dictate conducting statistical analyses on groups being compared. Since we make no comparisons between subQ and direct injection in the manuscript, beyond addressing the reviewer’s questions, we initially deemed additional statistical tests unnecessary. Our study’s focus is not on comparing transduction efficiencies of different injection methods. Nevertheless, we have now performed a t-test on the western blot quantifications in **Extended Data Fig. 5a**, which shows no significant differences between groups.

The IF images in **Extended Data Fig. 5b** and **5c** were derived from samples highlighted in green in **Extended Data Fig. 5a**, selected for their similar expression levels on western blot (also detailed in our figure legend). In our previous response, the 64% and 27% were simply the averages of FLAG signal in PLIN+ cells across the dorsal-ventral BAT sections shown in **Extended Data Fig. 5b** and **5c**. We simply stated these as a response to the reviewer, and these values are not in the manuscript, as they are not relevant to any conclusion or interpretation of our findings. We have also calculated the average a different way, where we divide the total FLAG OTSU area by total PLIN1 OTSU area from each individual IF image. These percentages work out to be 51% and 26%. Again, none of these values are in the paper, nor are any of these values pertinent to our interpretations. Thus, they are not included in our manuscript. Additional details of these analyses is available in the methods section and detailed in the figure legend.

All mice within each experiment, whether injected with L-GFP or L-CKB, were treated identically. Therefore, the method used (surgical or subcutaneous) has no impact on our interpretation of the differences between L-GFP and L-CKB.

Reviewer #3 (Remarks to the Author):

3.1. Based on the latest version of the paper, the claim that FCC is the predominant futile cycle is not supported by the data. This issue arises from the authors comparing two different mitochondrial types that have significantly different maximal capacities. Based on the data (5g), it appears that mitochondria lacking CKB have a maximal capacity that is only three times higher than the basal function, with only a slight increase observed following NA stimulation. Additionally, TNAP inhibition obstructs only 50% of this increase, which poses a concern given the high variation among the data points. Furthermore, the in vivo data (5j) does not align with the ²o data.

RESPONSE TO REVIEWER: We have now removed all mention of “predominant” in the manuscript. We are not comparing 2 different mitochondrial types to make quantitative claims about the FCC. Our quantitative claims regarding the FCC are based solely on TNAP inhibition data obtained from the same cells with the same mitochondrial type (cells rescued with

mitochondrial-specific CKB). The implication from the reviewer is that a discrepancy between the in vivo and in vitro data somehow invalidate our overall interpretations. However, our observations align with the literature such as studies on germline Ucp1 KO mice^{2,3}. In vivo experiments inherently involve multiple cell types responding to NA, which dampens observed differences. Isolated cells with the intended genetic modification provide a more focused and robust signal, as demonstrated in our study and prior work by the Cannon/Nedergaard group (**Reviewer only Fig 2**)³.

Reviewer only Fig 2. a, Norepinephrine (NE)-dependent thermogenic response between wild type, Ucp1(+/+), and Ucp1 knockout, UCP1(-/-) mice. Data taken from Fig. 2c of Feldmann *et al*, *Cell Metabolism*, 2009. **b**, NE-dependent thermogenic response of acutely isolated brown adipocytes from wild type, Ucp1(+/+), and Ucp1 knockout, UCP1(-/-) mice. Data taken from Fig. 1a of Matthias *et al*, *JBC*, 2000.

References

- 1 Himms-Hagen, J. Regulation of metabolic processes in brown adipose tissue in relation to nonshivering thermogenesis. *Adv Enzyme Regul* **8**, 131-151, doi:10.1016/0065-2571(70)90013-0 (1970).
- 2 Feldmann, H. M., Golozoubova, V., Cannon, B. & Nedergaard, J. UCP1 ablation induces obesity and abolishes diet-induced thermogenesis in mice exempt from thermal stress by living at thermoneutrality. *Cell Metab* **9**, 203-209, doi:10.1016/j.cmet.2008.12.014 (2009).
- 3 Matthias, A. *et al*. Thermogenic responses in brown fat cells are fully UCP1-dependent. UCP2 or UCP3 do not substitute for UCP1 in adrenergically or fatty acid-induced thermogenesis. *J Biol Chem* **275**, 25073-25081, doi:10.1074/jbc.M000547200 (2000).